# A Mechanistic Analysis of Low-Precision Instabilities in Microscaling Formats

**Chloe Huangyuan Su**[* 1,2]

**Mujin Kwun**[1]

**Stephanie Gil**[2]

**Sham Kakade**[1,2]

**Nikhil Anand**[* 1]

[1]**Kempner Institute for the Study of Natural and Artificial Intelligence, Harvard University**
[2]**Department of Computer Science, Harvard University**

**Reviewed on OpenReview:** `https://openreview.net/forum?id=I5bxWT7Xfw`

## Abstract

Training large language models is an expensive, compute-bound process that must be repeated as models scale, algorithms improve, and new data is collected. To address this, next-generation hardware accelerators increasingly support lower-precision arithmetic formats, such as the Microscaling (MX) formats introduced in NVIDIA's Blackwell architecture. These formats use a shared scale within blocks of parameters to extend representable range and perform forward/backward GEMM operations in reduced precision for efficiency gains. In this work, we investigate the challenges and viability of block-scaled precision formats during model training. Across nearly one thousand language models trained from scratch – spanning compute budgets from $2 \times 10^{17}$ to $4.8 \times 10^{19}$ FLOPs and sweeping over a broad range of weight–activation precision combinations – we consistently observe that training in MX formats exhibits sharp, stochastic instabilities in the loss, particularly at larger compute scales. To explain this phenomenon, we conduct controlled experiments and ablations on a smaller proxy model that exhibits similar behavior as the language model, sweeping across architectural settings, hyperparameters, and precision formats. These experiments motivate a simple model in which multiplicative gradient bias introduced by the quantization of layer-norm affine parameters and a small fraction of activations can trigger runaway divergence. Through *in situ* intervention experiments on our proxy model, we demonstrate that instabilities can be averted or delayed by modifying precision schemes mid-training. Guided by these findings, we evaluate stabilization strategies in the LLM setting and show that certain hybrid configurations recover performance competitive with full-precision training. We release our code as supplementary materials.

## 1 Introduction

Large language models (LLMs) have improved dramatically in recent years, largely by scaling their capacity and the quantity of training data (Kaplan et al., 2020; OpenAI, 2025; DeepMind, 2025; Anthropic, 2025; Grattafiori et al., 2024). For instance, training the Llama 3.1 405B model required more than $10^{25}$ FLOPs and

---

*Equal contribution.

utilized up to 16,000 H100 GPUs (Grattafiori et al., 2024). Scaling these models involves not only the initial, compute-intensive pretraining phase but also frequent retraining as new data, algorithms, or architectures emerge, as well as post-training protocols that prepare the model for inference/deployment.

To reduce these computational burdens, recent hardware advancements have introduced native support for lower-precision computations, such as FP8 training in NVIDIA H100 GPUs (Micikevicius et al., 2022a; Noune et al., 2022). Hardware accelerators powered by NVIDIA's Blackwell architecture further extend these capabilities with standardized, shared-scale Microscaling (MX) formats like MXFP8 and MXFP6 (NVIDIA, 2025). These formats store a per-block shared scale, which expands the effective dynamic range with minimal memory overhead, while simultaneously enabling GEMMs at lower precision (Rouhani et al., 2023; Darvish Rouhani et al., 2023b). While pretraining is typically done in 16 or 32-bit precision, some quantization schemes are already seeing industry adoption; for example, DeepSeek-V3 employs tile-wise FP8 quantization within large tensors (Liu et al., 2024), while Cohere's Command A model was trained in FP8 while reserving higher-precision operations for activation functions and attention mechanisms (Cohere et al., 2025). At an even larger scale, the Llama-4 series of models is reported to have been pretrained in FP8 precision across nearly 32,000 GPUs (Meta, 2025). On the deployment side, methods like QAT and mixed-precision fine-tuning further underscore the importance of understanding low-precision training dynamics (Jacob et al., 2017; Abdolrashidi et al., 2021; Shao et al., 2024).

Two primary challenges accompany the adoption of low-precision formats for training. First, there is a potential performance tradeoff, where reducing precision may result in degradation of loss and downstream accuracy, which can be characterized through scaling laws that account for both compute and precision (Kumar et al., 2024). Second, instabilities during training can occur, often manifesting as abrupt spikes in the loss curve that disrupt convergence (Fishman et al., 2024; Lee et al., 2025). When these instabilities push optimization into regions from which recovery is impossible, they obstruct our ability to extract valid scaling laws, making it impossible to even assess the tradeoffs introduced by low-precision training.

In this work, we set out to understand the training dynamics of low-precision MX precision formats to identify format prescriptions for language model training on next-generation hardware. However, like prior observations on (albeit non-MX) low-precision training by Fishman et al. (2024); Lee et al. (2025), we found that training frequently became unstable, particularly for larger, compute-intensive models. The instabilities are pervasive, emerging across a broad range of activation functions, model scales, quantization formats, and hyperparameter settings.

Because large-scale language model (LM) sweeps are computationally intensive and involve many entangled components, we turn to a controlled synthetic setting to understand the origin of these instabilities. Specifically, we present a residual multi-layer perceptron (MLP) model that captures key architectural components of the LM, and allows us to identify conditions under which training becomes unstable. In particular, we are able to perform hyperparamter sweeps, ablations across MX configurations, quantization schemes (e.g., forward-only vs. full quantization), and activation functions, and analyze their effects on stability.

Our findings support a phenomenological explanation in which training instabilities primarily arise from systematic bias in gradient estimates introduced by quantization. We find that the primary contribution to this bias is the quantization of the layer normalization (layernorm) affine weights, whose values often become tightly clustered over the course of training. When the values within a block converge too closely, division by the shared block scale can clamp all values in that block to the largest representable number, destabilizing training. This mechanism appears consistent with the failure modes observed in the LM setting, and we use this insight to design mitigations that stabilize LM training, including disabling layernorm quantization and using high precision in selective parts of the network computation.

*Authors' Note:* After this work was completed, several modifications to the original MX rounding algorithm have been proposed, such as in NVIDIA et al. (2025) and Cook et al. (2025). Our contribution is primarily isolating and quantifying a mechanism that makes certain rounding prescriptions unstable, and to offer guidance for designing and validating new ones.

## 2 Related Work

### 2.1 Low-Precision Instabilities

Training large Transformer models at scale can reveal instabilities that can disrupt or even halt learning (Liu et al., 2024; Chowdhery et al., 2022; Dehghani et al., 2023; Zhang et al., 2022; Molybog et al., 2023; Fishman et al., 2024; Zoph et al., 2022; Ma et al., 2025; Takase et al., 2025). In some cases, these issues are exacerbated or directly triggered by low-precision quantization. For example, Fishman et al. (2024) demonstrate that FP8 pretraining becomes unstable when combined with the SwiGLU activation function, attributing the issue to an outlier amplification effect that worsens due to progressive weight alignment over the course of training. Similarly, Lee et al. (2025) report that approximately 10% of BF16 runs using the NanoGPT codebase fail to converge, whereas full-precision (TF32) training exhibits no such failures. Other works (Sun et al., 2024; Bondarenko et al., 2023; Xu et al., 2023), point to activation outliers and gradient norm growth as contributors to these failures while Tseng et al. (2025) proposes a stochastic rounding based algorithm to stabilize training in MXFP4 formats. Meanwhile, DeepSeek-V3 also attributes certain training failures due to blockwise quantization of activation gradients (Liu et al., 2024), underscoring the breadth of challenges introduced by quantization schemes. Wortsman et al. (2024) use small-scale proxy models to study training instabilities in the context of growth of output and layer logits. We adopt a similar approach, and use a simplified proxy model to understand the origin of low-precision instabilities in LLMs.

### 2.2 Review of MX Formats and Experimental Approach

MX formats are a class of low-precision numerical representations designed to enhance the efficiency of deep learning models (Darvish Rouhani et al., 2023a; Rouhani et al., 2023). We defer a detailed review of the MX scheme to Appendix A. To summarize, we represent a block of $k$ values, $\{V_i\}_{i=1}^k$, using a single shared scale factor $X$ and $k$ corresponding low-precision elements $\{P_i\}$ where the $P_i$ are obtained by casting $V_i/X$ to the specified low-precision format. We present results for a block size $k = 32$ to match what will be hardware supported. The scale $X$ is calculated using $X = 2^{\lfloor \log_2(\max_i(|V_i|)) \rfloor - e_{\max \text{ elem}}}$ where $e_{\max \text{ elem}}$ is the exponent of the largest normal number representable in the chosen element data format.

In our experiments, we quantize both weights and activations using these MX formats using the MX Pytorch Emulation Library (Microsoft, 2024). As described in Appendix A, this quantization is applied dynamically to the inputs of matrix multiplication operations. Our experiments primarily involve the original version of the MX rounding algorithm as proposed in Darvish Rouhani et al. (2023a), although we experiment some variations in Section 5.2. On the systems side, NVIDIA Transformer Engine exposes optimized Transformer kernels that support FP8 training via dynamic scaling and `amax`-tracking, and more recently discusses MXFP8-style block scaling where scaling granularity is reduced from per-tensor (FP8) to per-block (MXFP8) to better preserve dynamic range under aggressive quantization (NVIDIA, 2024; 2025). While our study uses the MX PyTorch Emulation Library to isolate algorithmic effects independent of any specific kernel implementation (Microsoft, 2024), these TE mechanisms provide a complementary, hardware-oriented perspective on how scaling granularity and format selection influence stability.

## 3 LLM Experiments

### 3.1 Setup

For our LM experiments, we use OLMo (Groeneveld et al., 2024) combined with the MX PyTorch Emulation Library (Microsoft, 2024) to enable training under various low-precision configurations. All language models use the GeLU activation function; full hyperparameter details are provided in Table 5. We sweep over a wide range of MX precision formats for both weights and activations, including two FP6 variants (`E3M2`, `E2M3`), two FP8 variants (`E4M3`, `E5M2`), and a bfloat16 baseline. Each configuration applies full quantization to both forward and backward passes to both weights and activations, as implemented in the Microscaling library (Microsoft, 2024). For each format, we train approximately 70 models[1] spanning compute budgets

---

[1]Some runs crashed and could not always be resumed, leading to small differences in number of models trained for each format.

from $2 \times 10^{17}$ to $4 \times 10^{19}$ FLOPs. Model sizes range from ∼20M to ∼1.7B parameters. Token counts are determined using an adapted version of the FLOP accounting code from Brandfonbrener et al. (2024), originally developed for OLMo scaling law experiments. Token-to-parameter ratios in our sweep range from approximately 2 to 156. Models are trained on the Fineweb-Edu dataset Penedo et al. (2024) and the StarCoder dataset Li et al. (2023), with the longest runs trained on 35B tokens and the shortest runs corresponding to models trained on 301M tokens.

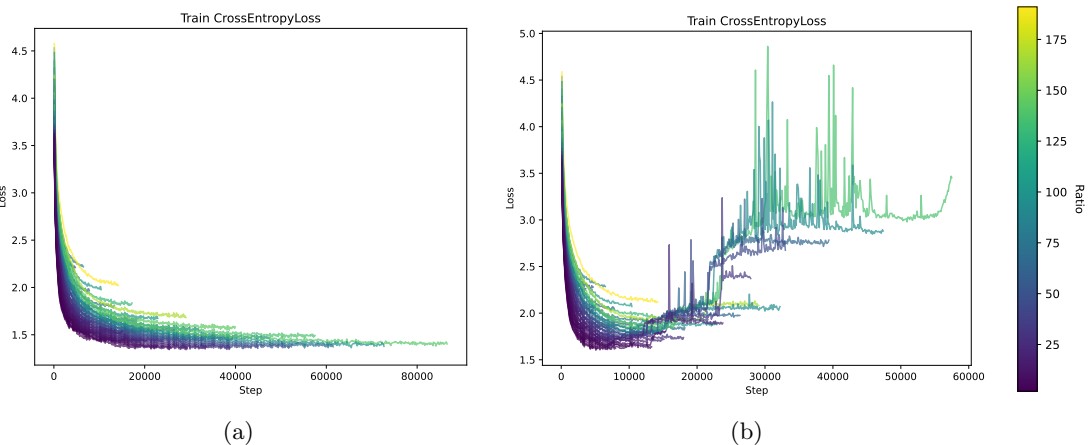

Figure 1: OLMo Train loss on Fineweb-Edu for weights and activations in `bf16-bf16` (left) and MXFP8 `E5M2-E5M2` (right) for various FLOP budgets, for the same hyperparameter configuration. Some runs, particularly larger models that are trained for longer, become unstable and never recover. Low precision computations are done in both forward and backward steps. Color bar on the right shows the token-to-parameter ratio.

## 3.2 Instabilities in Low Precision

Figure 1a shows the training loss and gradient norm trajectories for bfloat16 models. Training remains stable, with smooth convergence. By contrast, Figure 1b illustrates example instabilities in the MXFP8 `E5M2-E5M2` weights-activations configuration, where some training runs exhibit sharp upward spikes in loss and large increases in gradient norm magnitude. We find these instabilities to be common across other low-precision MX configurations and hyperparameter settings, as noted in Appendix J. We observe the instabilities mainly in larger, longer-trained models and that importantly, when training is destabilized, training does not recover, and the loss continues to diverge. While the loss spikes appear abruptly, the gradient norm typically grows more gradually (see, e.g., examples in Appendix J) and fails to decrease over time as seen in stable bfloat16 training. This behavior strongly suggests biased gradient estimates, a point that we will investigate further in subsequent sections.

## 4 Synthetic Experiments

### 4.1 Setup

Our LM experiments with OLMo involve many potentially interacting components, and it is computationally expensive to determine exactly where the low-precision failure mode occurs. To facilitate this task, following Wortsman et al. (2024), we develop a small-scale proxy model. Given an input $x \equiv A_0 \in \mathbb{R}^{d_{\text{model}}}$, we consider a network composed of $L$ residual layers indexed by $k = 0, \ldots, L-1$. The hidden state at each layer is computed as:

$$h_k = \mathbf{W}_k^{(1)} \text{LN}(A_{k-1}), \quad A_k = A_{k-1} + \mathbf{W}_k^{(2)} \phi(h_k), \tag{1}$$

where LN denotes layer normalization and $\phi$ is the activation function (e.g., ReLU, GeLU, SwiGLU). Each residual block contains two weight matrices: $\mathbf{W}_k^{(1)}$ projects to the hidden dimension, and $\mathbf{W}_k^{(2)}$ projects back to $d_{\text{model}}$. By default, the hidden size is set to $4d_{\text{model}}$[2])

This student/proxy model is only useful insofar as it (at least partially) mimics the failure modes of the LM setting, so let us note the simplifications performed on the language model in order to obtain the proxy model. First, we dispense with the self-attention blocks since ablating over attention did not change the qualitiative nature of the divergences we observed. Second, we remove the embedding layers since our goal is to understand exactly how low-precision block scaled arithmetic biases gradient computations, as well as simplify the various types of LM layernorms (such as $QK$-norms) into a single layernorm. Finally, we also train with MSE loss rather than cross-entropy, although we experimented with a distributional KL loss and again did not observe qualitative differences. While we show that this model nevertheless remains instructive and predictive of the mechanistic origins of the LM instabilities, we caution that stability in this minimal model as a necessary (though perhaps not sufficient) condition for stability in the full LM. Appendix D inludes more experiments on how some of these simplifications affect the training dynamics of the model.

The targets are generated by a fixed auxiliary/teacher model that serves as a sufficiently complex learnable function (Lin et al., 2025), and whose architecture can be taken to be the same as the student's without the layer normalization. For sweeps where we change the depth and width of the student, we similarly scale the teacher model. The inputs $x$ are drawn i.i.d. from a standard Gaussian, without cycling, using a fixed seed to ensure consistent batch order.

To isolate the effect of precision, we train two copies of the student model from the same initialization. The first is trained in full precision (FP32). After training, the weights are reset to their initial state and retrained using a low-precision MX format, with quantization applied to both forward and backward passes as described in Section 2.2. Because the random seed, kernel determinism, initialization, data, and batch order are identical, any behavioral difference is attributable mainly to the change in precision.

**Hyperparameter choices**  A key point explicated in Appendix C is that there are hyperparameter choices for which the model in Equation (1) will give rise to train instabilities (even in FP32 precision). This is not necessarily a precision issue, but rather due to the fact that in any SGD method there exists some small probability of taking wrong gradient step(s). If the size of the steps are large due to, e.g., a large learning rate, this will be visible as a sudden spike(s) in the loss. In order to move away from these "expected" instabilities, before ablating or changing various components of the architecture, we carefully tune hyperparameters for each depth and width configuration in which all high-precision runs are stable, but low precision is not (at least for a canonical choice of activation function such as GeLU). For the same reason, we fix a moderately large batch size (2048) throughout to reduce variance in gradient estimates.

## 4.2  The Effect of Activation Functions and layernorms

Having fixed a hyperparameter regime in which instabilities only appear in low precision, we first ablate the choice of activation function and the inclusion of layer normalization. In Equation (1), this corresponds to varying $\phi(\cdot)$ and including the presence of $\text{LN}(\cdot)$.

In Figure 2a, we observe that with layer normalization enabled, both GeLU and SwiGLU activations exhibit instability in low precision, with SwiGLU being significantly more prone to divergence. This is consistent with the findings of Fishman et al. (2024), though our results show that SwiGLU also destabilizes training in high precision, suggesting that it generally increases stochasticity at least for this particular choice of hyperparameters, though these instabilities are generally recoverable in high precision. We observe two irrecoverable instabilities in GeLU under low precision that are absent in high precision.

Next, we look at the inclusion of layernorm. In Figure 2b, we observe that the loss improves with the removal of layernorm. This is expected as the teacher network does not contain a layernorm so that student model is able to more accurately represent its outputs. However, removing layernorm tends to stabilize low-precision

---

[2]In the case of SwiGLU, following Shazeer (2020) we reduce the hidden dimension from $4d_{\text{model}}$ to $\frac{8}{3}d_{\text{model}}$ to maintain parity in parameter count.

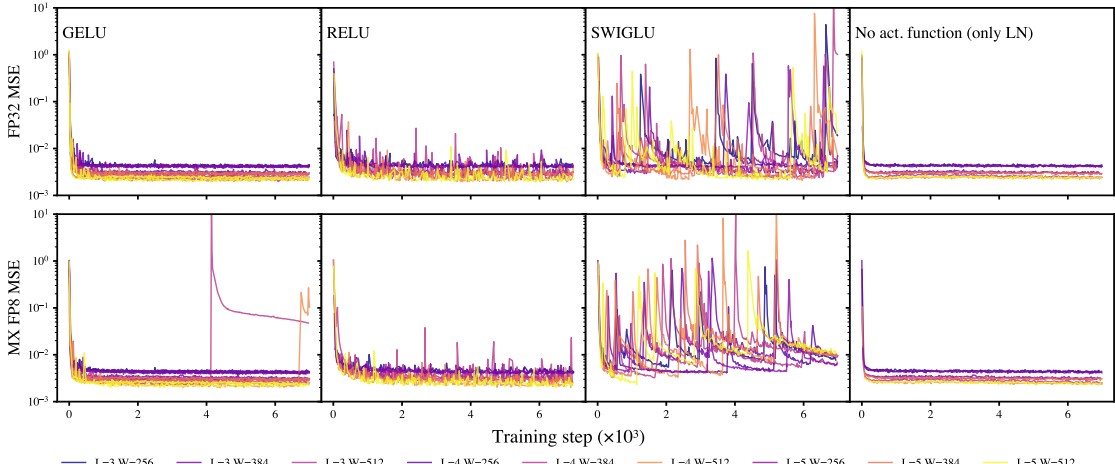

(a) **(Proxy Model)** Loss curves of different activation functions with the inclusion of layernorm, for various model depth/width settings. With layer normalization enabled, both GeLU and SwiGLU activations exhibit instability in low precision for some configurations, with SwiGLU being significantly more prone to divergence, though we note that in high precision these divergences are often recoverable.

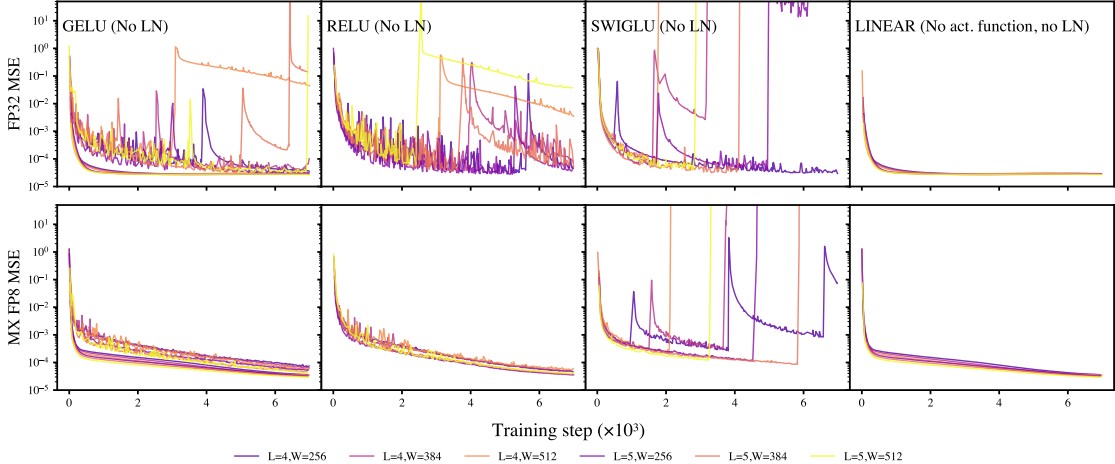

(b) **(Proxy Model)** Loss curves of different activation functions without layernorm. When layernorm is removed, lower precision runs tend to become more stable.

Figure 2: Shows the comparison between full and low precision training across different activation functions, with and without layernorm.

training runs and destabilize high precision runs (for the same choice of hyperparameters in Figure 2a). At first glance, these results are perplexing since it appears that *low precision is more robust to removal of layernorms.* We will return to this point in Section 5 when we explicate the subtleties of layernorms in block scaling formats.

## 5 Overflow Dynamics

Typically, instabilities in low precision happen due to over/underflow issues that can bias the gradient. However, in a block scaling format, it is unclear how gradient bias can accumulate when the shared scale explicitly puts nearly all values within a representable range.

### 5.1 Overflow Issues with layernorms

To understand this, we begin by examining a concrete example of MXFP8 `E4M3` as specified in Darvish Rouhani et al. (2023a). The left panel of Fig. 3 plots the relative gap $(x_{t+1} - x_t)/x_t$ between successive *positive* codes in this format, ordered from index 0 (the smallest sub-normal, $2^{-9}$) up to index 125 (448). The index stops at 125, rather than the expected $2^7 - 1 = 127$, because S 1111 111$_2$ is reserved for the NaN symbol, which would otherwise correspond to a value of 480, and S 0000 000$_2$ is the zero code, leaving 126 remaining codes (Darvish Rouhani et al., 2023a). We can note the following:

**1.** For a fixed exponent bin the relative gap starts at 12.5% and decays to 6.6% as the mantissa increases.

**2.** There is an overflow region (left of Figure 3) when the value exceed the largest representable normal number (448). Typically, these values are clamped down to 448.

The latter observation above means that if a block of values lies within a sufficiently small band, these values may end up in the gray overflow region of Figure 3 after dividing by the block scale. For example, from Algorithm 1, for the case of MXFP8 `E4M3` which has $e_{\max}^{\mathrm{elem}} = 8$ the overflow criteria for a given value $v$ within a block with a shared scale $X$ is

$$\left| \frac{v}{X} \right| > 448 \quad \Rightarrow |v| > 0.875 \times (\text{abs. max within block}). \tag{2}$$

This type of overflow region was noted for the case of narrower MXFP4 format in Tseng et al. (2025). We show that, while MXFP8 E4M3 has a larger dynamic range, the same effect becomes consequential in practice because layernorm *affine* weights are tightly clustered and particularly susceptible to having *all* values within a block falling in this range. For example, layernorm weights typically follow log-normal distributions with scale $e^\mu \sim 1$ and deviation $\sigma \ll 1$, and so a block of weights might look something like

```
[0.89740956, 0.89628334, 0.88358812, 0.88474816, 0.90372837  ...]
```

which all end up in the overflow region of Figure 3 after dividing by $X = 2^{\lfloor \log_2(\text{abs. max}) \rfloor - e_{\max}^{\mathrm{elem}}} = 2^{-8}$. In our experiments, the impact of this effect is shown in the middle plot of Figure 3. In the proxy model setting, in some cases, nearly all of the layer norm weights fall within the band required to flow into the last bucket, losing heterogeneity in nearly all blocks when they are clamped to the maximum normal value after scale division. Note that this explains, at least partially, why removing the layernorms stabilized low-precision training in Figure 2b. While a different format, like MXFP8 `E5M2` may avoid this issue, the loss of precision from having only two mantissa bits can still lead to training instabilities.

In a typical LLM setting such as OLMo, several different types of layernorms experience different degrees of clamping to the last quantization bin. As seen in the middle-top plot of Figure 3, some components such as the attention layernorms, remain relatively well behaved throughout training, whereas others, like the FFN layernorms or the $QK$ layernorms (Henry et al., 2020), can experience large, sudden overflow issues for nearly 75% of weights[3]. While it's possible to disable the affine transformation of layernorms in the LM setting and we indeed find that this significantly enlarges the stability window (see Appendix F), we also observe that some residual instability still remains at larger training durations, perhaps due to the presence of this effect in a small fraction of the activation values. More broadly, this finding indicates a problem with applying shared-scales to blocks of weights that follow approximately log-normal distributions, which may not have a well-defined notion of a "max" relative to a resolution fixed by a given precision scheme. A scale that adapts to both *min* and *max* might avoid the bias; we defer this to future work and note the prescription proposed in Mishra et al. (2025) as a potential solution. On the activation side, we find that this effect is apparent in roughly $\sim$1% of values in our synthetic experiments and $\sim$0.5% of values in OLMo (shown in the right subplot of Figure 3).

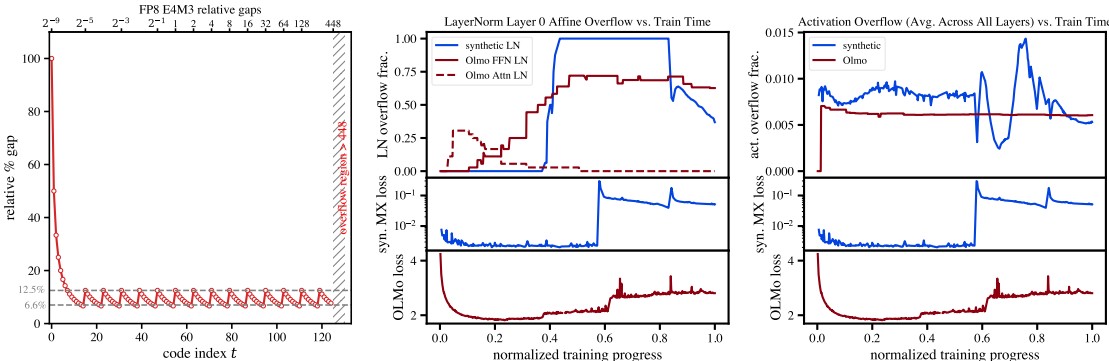

Figure 3: **Proxy and LLM LayerNorm Overflow**. **Left**: relative gap $(x_{t+1}-x_t)/x_t$ for successive positive FP8 `E4M3` codes (sign bit stripped). Within each exponent band the gap decays from 12.5% to 6.6%; the hatched region marks values that would be clamped once the scaled magnitude exceeds the representable limit of 448. **Center**: Top subplot shows what fraction of layernorm affine parameters end up in the last quantization bin after division of the shared scale in the first layer of the network. For OLMo, we look at the FFN layernorm and the attention layernorm. The synthetic loss in this case exhibits a divergence in MX precision (but is stable in FP32 precision), and corresponds to the student-teacher setup of Equation (1) with four layers and $d_{\mathrm{model}} = 512$ and $\eta = 6 \times 10^{-4}$. **Right**: Shows the fraction of activation values (averaged across layers) that end up in the last quantization bin after division by the shared scale.

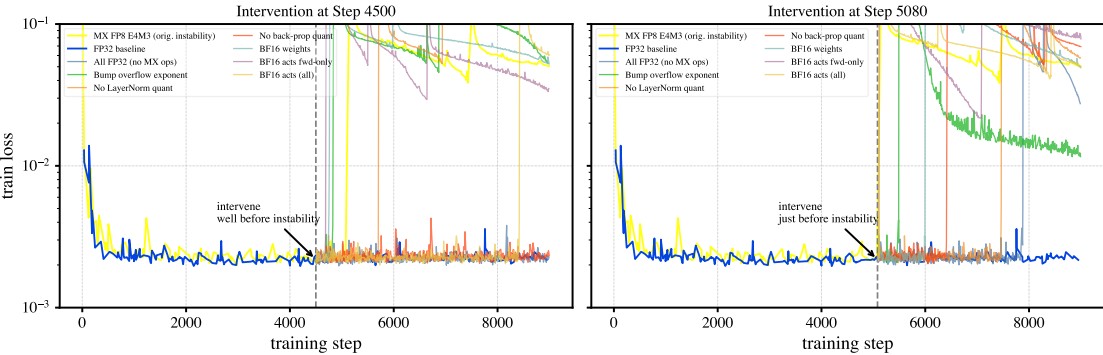

Figure 4: **(Proxy Model)** Intervention experiment for a synthetic student-teacher model with $d_{\mathrm{model}}=512$, four layers, and learning rate $\eta=6\times10^{-4}$. Training is stable in FP32 (blue) but diverges in MXFP8 `E4M3` (yellow) around step 5100. We test two intervention timings: step 4500 (left, well before instability) and step 5080 (right, just before instability). Early interventions, like disabling backward-pass quantization or switching to high-precision (FP32), successfully prevent divergence, while using high precision for the activations (bfloat16) can greatly delay it. Late interventions cannot avert instability but can only delay it; the most effective are switching to FP32 or skipping quantization of layernorm weights.

## 5.2 Potential Mitigations

To clearly establish causality of which components can (de)stabilize training, we ask whether an impending divergence can be averted by *in-situ* interventions to the training recipe. Figure 4 tracks a configuration that is stable in FP32 but diverges in MXFP8 `E4M3`. This setting corresponds to the previously described student-teacher scenario with four layers and model dimension $d_{\mathrm{model}} = 512$. The instability starts approximately at step 5090 and we consider interventions just before the instability, at step 5080, and well before the instability, at step 4500. For each intervention we keep the random seed, model state, and batch sequence identical, so

---

[3]We observe similar effects when LayerNorm is replaced with RMSNorm; see Figure 19.

the training state at the intervention step is the same as in the baseline run, so any divergence afterward is therefore solely attributable to the intervention.

**Switching entirely to FP32 precision for remaining training steps.** Intervening with FP32 significantly stabilizes training if the change is made sufficiently early (step 4500), but it is ineffective if applied immediately before instability (step 5080). However, even at the later intervention, FP32 prolongs training stability more effectively than the other approaches.

**Increasing the shared exponent by one (bump exponent).** Adjusting the exponent to avoid the last bucket overflow for blocks that have values that fall into the range in Equation (2) does not mitigate instability, which may be due to insufficient precision improvement from a single increment too late in training.

**Avoiding MX quantization for layernorm affine parameters.** Intervening by omitting quantization of layernorm parameters partially stabilizes training and delays instability significantly at both intervention steps, indicating that layernorm parameters do contribute to instability dynamics. However, eventual instability suggests a residual effect from quantized activations.

**Precision adjustments in forward and backward passes.** We explored quantizing weights and activations only during the forward pass (no backward-pass quantization); maintaining weights in bfloat16 and activations in MXFP8 (both passes); maintaining activations in bfloat16 for the forward pass but MXFP8 for backward (with MXFP8 weights); using BF16 activations for both forward and backward passes while quantizing weights with MXFP8. As seen in Figure 4, among these, applying the intervention just before instability (step 5080), bfloat16 activation precision in both passes consistently provides the strongest immediate stabilization, closely followed by disabling backward-pass quantization. When interventions occur earlier (step 4500), not quantizing the backward step performs comparably to the FP32 baseline, while fully bfloat16 activations delay instability considerably yet eventually become unstable. These results suggest a stochastic model in which multiple interacting factors can cause gradient bias/influence instability likelihood.

**Key Takeaways** The dominant MX precision-specific bias comes from overflow of clustered layer-norm affine weights (and a small fraction of activations). Our intervention experiments show that raising precision in key parts of the computation, such as increasing the precision of layer norms or activations, can greatly improve stability.

## 6   Scaling Law Fits and Loss Curves after Mitigation

| Weight | Activation | $D/N$ **Ratio** | | | | | | |
|---|---|---|---|---|---|---|---|---|
| | | **140.96**
$N$=0.16B | **99.19**
$N$=0.19B | **70.91**
$N$=0.23B | **37.86**
$N$=0.31B | **21.28**
$N$=0.42B | **16.23**
$N$=0.48B | **12.51**
$N$=0.54B |
| bfloat16 | bfloat16 | 0.710 | 0.703 | 0.698 | 0.691 | 0.688 | 0.686 | 0.686 |
| MXFP8 E4M3 | bfloat16 | 0.0 | -0.002 | -0.002 | 0.0 | 0.0 | 0.0 | 0.0 |
| MXFP8 E5M2 | bfloat16 | 0.105 | 0.107 | 0.112 | 0.004 | 0.002 | -0.001 | -0.001 |
| MXFP8 E4M3 | MXFP8 E4M3 | 0.005 | 0.002 | 0.002 | 0.004 | 0.002 | -0.001 | -0.001 |
| MXFP8 E5M2 | MXFP8 E5M2 | 0.010 | 0.012 | 0.057 | 0.019 | 0.007 | 0.004 | 0.004 |

Table 1: **LLM Validation Losses:** The validation loss on Fineweb-Edu of high precision runs versus low precision with mitigations applied (values are shown as differences with respect to bf16-bf16 baseline; lower is better). For the last two rows, we quantize only the forward pass.

In addition to Figure 5 we provide scaling law for the mitigation where we quantize only the forward pass; this is shown in Figure 6 which can be compared against the bfloat16 baseline in Figure 7. Scaling law fits were performed using the methods described in Hoffmann et al. (2022); Brandfonbrener et al. (2024) where the validation loss was fit with a functional form

$$L(N, D) = E + \frac{A}{N^\alpha} + \frac{B}{D^\beta}, \tag{3}$$

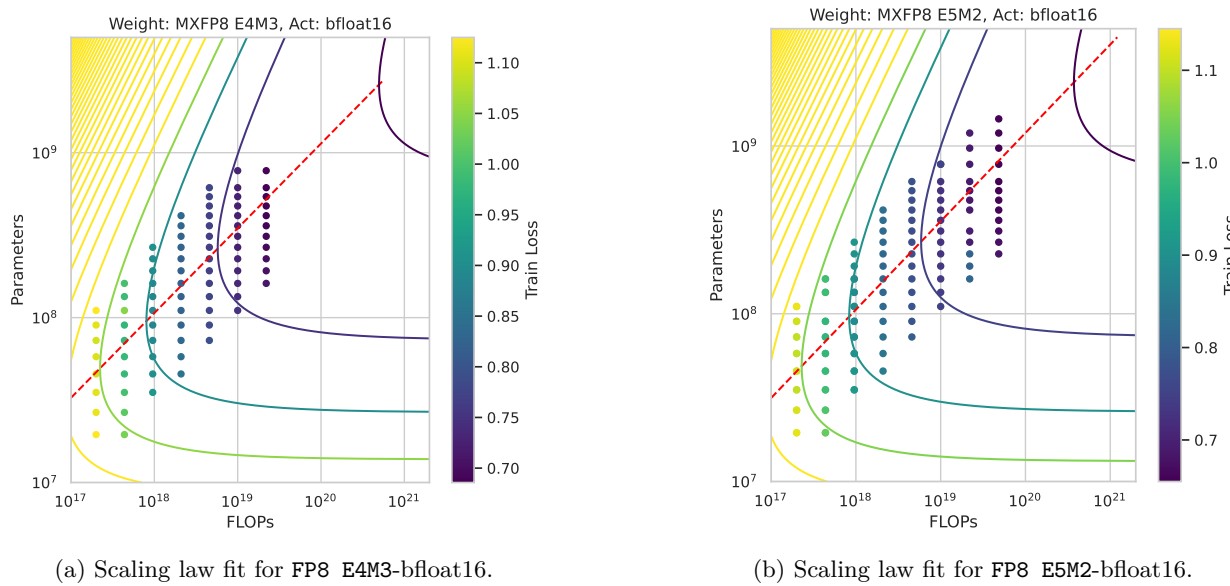

(a) Scaling law fit for `FP8 E4M3`-bfloat16.

(b) Scaling law fit for `FP8 E5M2`-bfloat16.

Figure 5: **LLM Scaling Laws:** Scaling law fit for combinations of precision formats of weights and keep the activations in high precision. Fit was calculated using a Chinchilla model for the loss; details and fit parameters are in Section 6.

for constants $A$, $B$, $E$, $\alpha$, and $\beta$. The fitted values of these constants are given in Table 2. We also provide the loss curves after implementing these mitigation strategies; these are shown in Figure 8 through Figure 11.

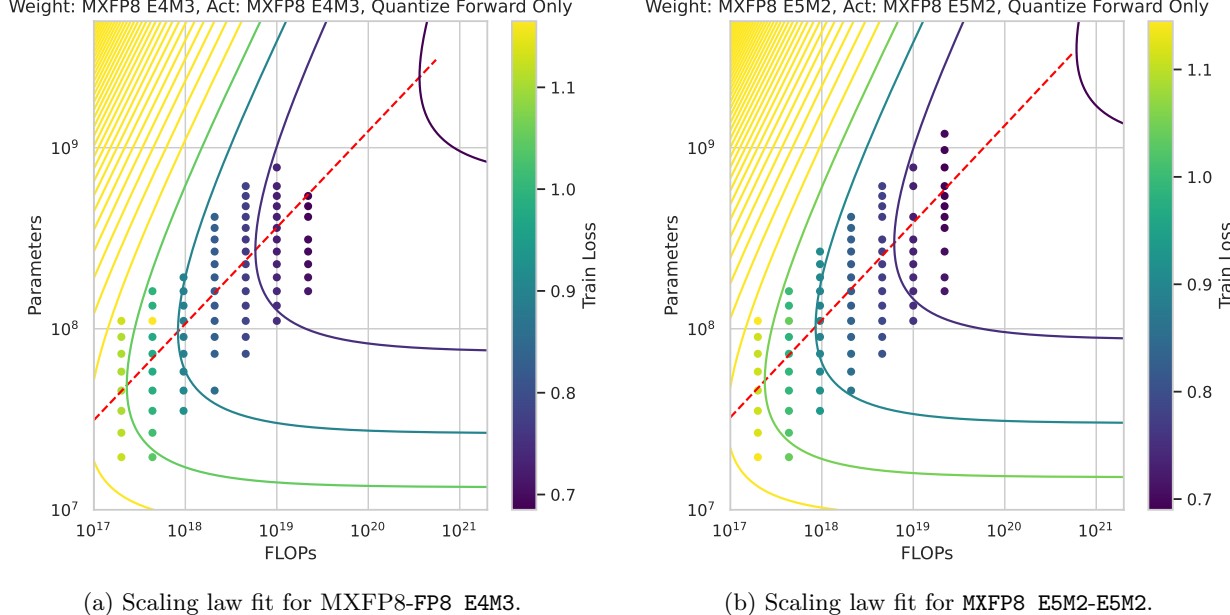

(a) Scaling law fit for MXFP8-`FP8 E4M3`.

(b) Scaling law fit for `MXFP8 E5M2`-E5M2.

Figure 6: **LLM Scaling Laws:** Scaling law fits for fixed stable of precision formats of weights and activations quantizing only the forward pass.

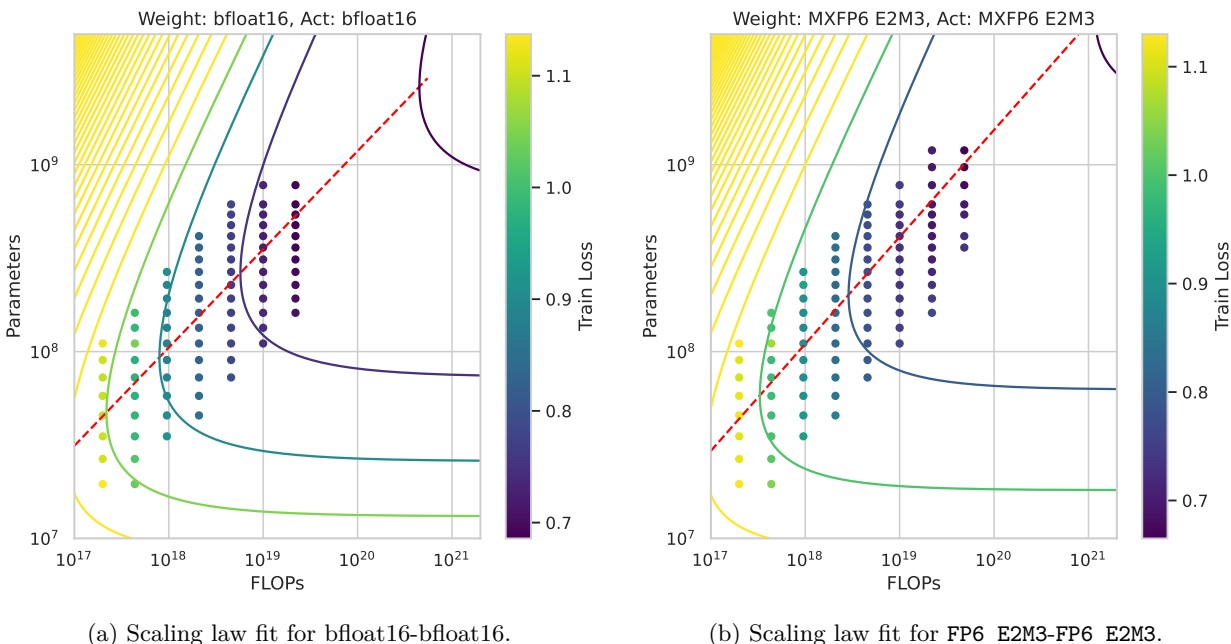

(a) Scaling law fit for bfloat16-bfloat16.

(b) Scaling law fit for `FP6 E2M3-FP6 E2M3`.

Figure 7: **LLM Scaling Laws:** Scaling law fits for bfloat16-bfloat16 (baseline) and for MXFP6 format.

| Weight | Activation | A | B | E | $\alpha$ | $\beta$ | $a$ |
|--------|-----------|-----|-----|-----|-----|-----|-----|
| `MXFP6 E2M3` | bfloat16 | 1.84e+03 | 8.77e+03 | 0.52 | 0.50 | 0.51 | 0.51 |
| `MXFP8 E4M3` | bfloat16 | 2.82e+03 | 2.04e+04 | 0.54 | 0.52 | 0.55 | 0.51 |
| `MXFP8 E5M2` | bfloat16 | 1.68e+03 | 1.84e+04 | 0.52 | 0.49 | 0.55 | 0.53 |
| bfloat16 | bfloat16 | 1.94e+03 | 2.18e+04 | 0.53 | 0.50 | 0.56 | 0.53 |
| `MXFP8 E4M3` | `MXFP8 E4M3` | 1.57e+03 | 2.11e+04 | 0.52 | 0.49 | 0.55 | 0.53 |
| `MXFP8 E5M2` | `MXFP8 E5M2` | 2.20e+03 | 3.98e+04 | 0.54 | 0.51 | 0.59 | 0.54 |

Table 2: **LLM Fitted scaling law parameters**: For the last two rows, we quantize only the forward pass. The last column is equal to the ratio $a = \beta/(\alpha + \beta)$, the exponent of the optimal model size relative to FLOPs.

# 7 Conclusion

We showed that training LLMs in shared-scale/MX configurations can lead to sharp, unrecoverable instabilities. Using large-scale LLM sweeps and a simple proxy model trained on synthetic data, we isolate a failure mode of quantization-induced gradient bias, where shared-scale clamping (particularly of layer-norm affine weights and to a lesser extent, other activations) injects gradient noise that ultimately destabilizes training. We evaluated several diagnostic mitigations, and found that stability can be preserved using higher precision in selective parts of the network computation. Looking ahead, continued hardware advances will expand the frontier of what is computationally feasible. Some concrete directions include: extending our proxy model to include mixture-of-experts with many layers, and other transformer-specific components to better predict instabilities; developing a clear theoretical picture of instabilities in optimization (see Appendix B); and designing new blockwise scaling schemes such as in Mishra et al. (2025) that adapt to skewed or tightly clustered distributions.

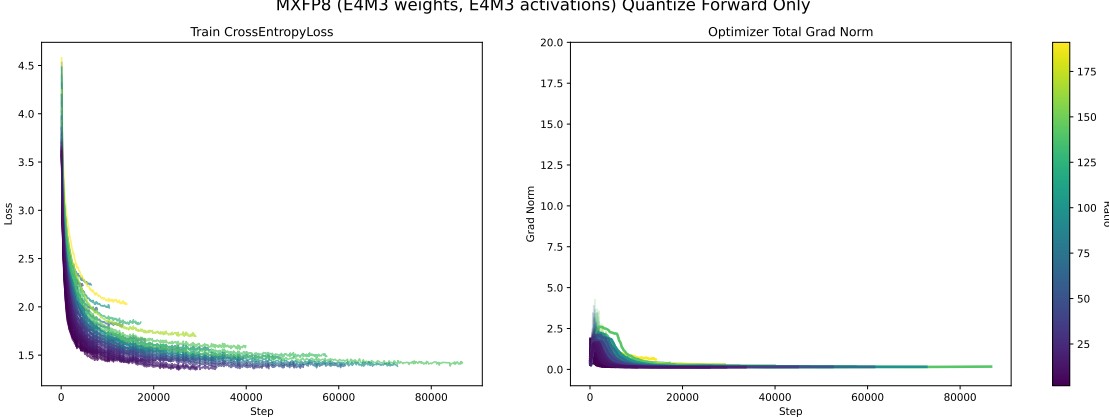

Figure 8: **LLM train loss and gradient norm** of `MXFP8 E4M3-MXFP8 E4M3` when quantizing only the forward pass.

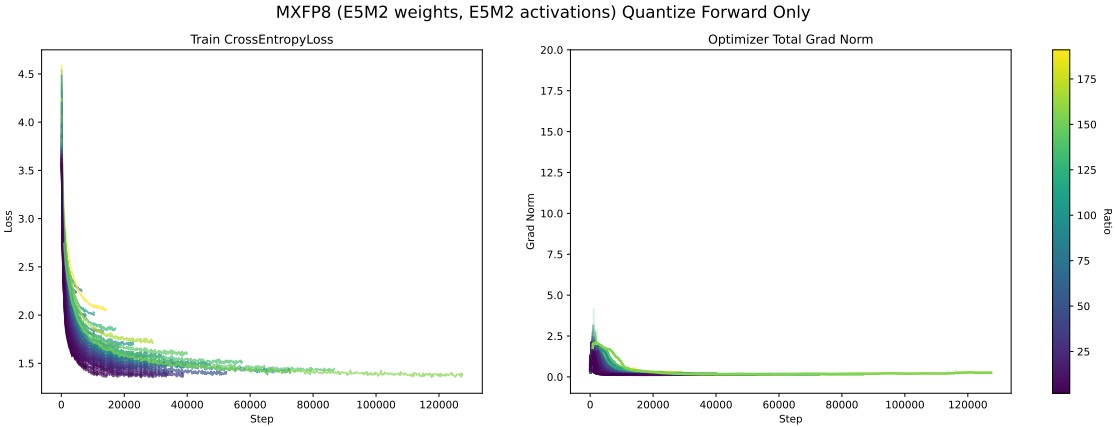

Figure 9: **LLM train loss and gradient norm** of `MXFP8 E5M2-MXFP8 E5M2` when quantizing only the forward pass.

## 8 Limitations

In this work, we provide insights into when LLM training instabilities arise using a proxy model. Specifically, we identify sources of instability, such as hyperparameters, that affect both high- and low-precision regimes. We also highlight instability factors specific to low-precision training, including sensitivity to learning rate, model depth and width, and the choice of activation function. We explore how different interventions stabilize training and enable us to fit precision-aware scaling laws, capturing how validation performance scales with model size and number of training tokens.

However, there are many interactions in model training and we cannot exhaustively cover them all in our sweeps. First, there are mitigation strategies not evaluated in this work, such as those that involve changing the quantization algorithm itself. We emphasize, however, that a core contribution of this paper is to explore a common failure mode in block-scaling formats and propose simple models to help reason about complex low-precision dynamics, rather than prescribe a single universal fix. Our results are primarily derived from decoder-only language models and residual MLPs; future work is needed to determine whether similar behaviors arise in MoE architectures. Finally, while we emulate MX formats in PyTorch, all experiments are conducted in software. Real-world deployment on Blackwell-class hardware may introduce additional sources of error due to rounding behavior, memory layout, or fused kernel execution not captured by our implementation.

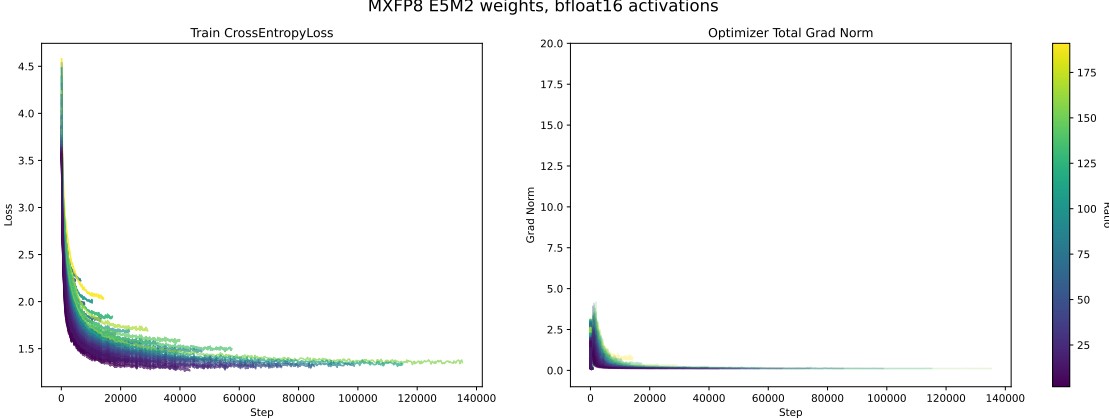

Figure 10: **LLM train loss and gradient norm** of `MXFP8 E5M2-MXFP8 E5M2` when activations are kept in bfloat16.

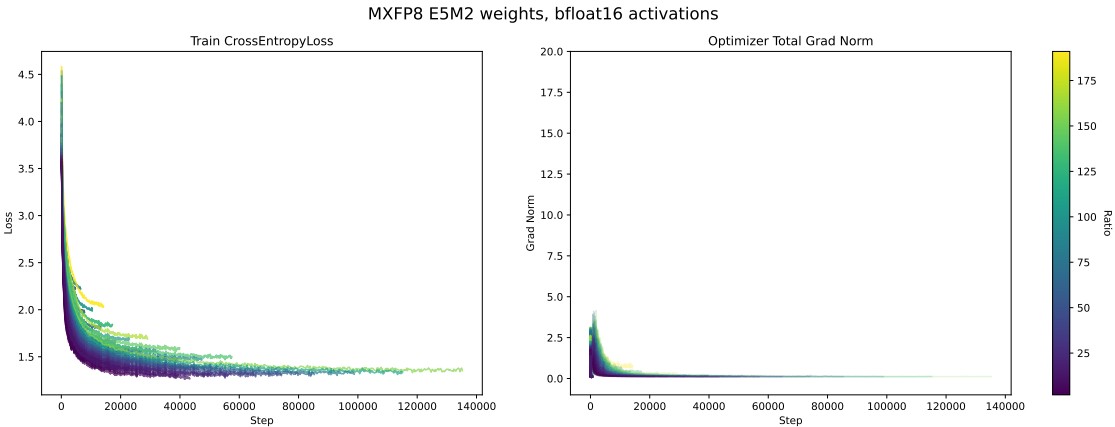

Figure 11: **LLM train loss and gradient norm** of `MXFP8 E5M2-MXFP8 E5M2` when activations are kept in bfloat16.

Another point is that we acknowledge as a limitation that our LM sweep uses GeLU rather than the SwiGLU activation now common in LMs. The proxy results in Figure 2b suggest SwiGLU is more failure-prone in low precision than GeLU or ReLU (removing LayerNorm stabilizes it only partially), so roughly speaking we would expect the LM to inherit the same effect. A full characterization, including whether the same clustering / overflow mechanism applies to the SwiGLU gating values themselves (and to the SiLU factor that multiplies them), is an interesting direction for future work.

Regarding our scaling law fits, we note that a small number of runs ($O(1)$ per format out of ~70) failed due to transient cluster issues unrelated to the precision format itself and were excluded from the scaling-law fits. This exclusion could in principle bias the fitted exponents, though we did not observe the crashes to be concentrated in any particular format. We emphasize that the scaling-law fits reported here are intended as a sanity check that our hybrid configurations recover bfloat16-level scaling, rather than as precise estimates of the exponents. A more careful analysis of block-scaled precision scaling laws, with confidence intervals on the fitted parameters, is an important direction for future work.

We also performed most of our experiments on relatively small models; our synthetic models run on NVIDIA H100s in a matter of minutes, and the largest language models used 16 H100s per run. While we generally expect the same effect to persist with scale, we cannot rule out that mitigations we proposed here will continue to work at larger scale. In particular, a systematic study of how block-scaling algorithm choice

interacts with model size and whether keeping a small set of sensitive computations in high precision remains sufficient to stabilize training when the bulk of forward and backward GEMMs are in MX low precision at $\geq$ 1B parameters, is an interesting direction for future work.

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

## A    Review of Shared-Scale Quantization

In this section we provide a self-contained review of block scaling quantization schemes, largely following Rouhani et al. (2023); Darvish Rouhani et al. (2023a). Taking a step back, the idea in shared-scale quantization methods is to introduce a number which represents the shared scale among a group of values that could, e.g., represent weights or activations. The idea is that low-precision data types tend to have a small representable range and quantization can clip very large values or zero-out smaller values. By dividing by the shared scale, the goal is to put these numbers in a representable range and save the scale such that it may be multiplied at the end of the computation. There are many choices for how to pick the scale, with pros and cons for each. For example, one approach is to have a single scale factor for the entire tensor, which has a very low memory overhead but is usually too coarse-grained and can lead to saturation issues. On the opposite end, one could keep a scale factor for every value in the tensor which obviously allows for higher accuracy but involves much more memory. Other approaches include tilewise scaling, where a scale factor is used for a fixed-size submatrix. This was the approach taken in Liu et al. (2024). In this work, we focus on *block* scaling methods, where a single 1-dimensional block of values shares a scale. In particular, we focus on the "microscaling" (MX) format, where each block consists of 32 values, with a shared scale that can be computed using Algorithm 1. When performing matrix multiplications or dot products, these shared scales are carried around and multiplied at the end of the computation (see Darvish Rouhani et al. (2023a) for the exact specifications).

---

**Algorithm 1** Convert $\mathbf{V} \in \texttt{HP\_DTYPE}^k$ to an MX block $\{X,\ P \in \texttt{LP\_DTYPE}^k\}$

---

**Require:** $k = 32$ (hardware block size),
  1: $e_{\max}^{\text{elem}}$ — exponent of the largest normal value in $\texttt{LP\_DTYPE}$
**Ensure:** Scale factor $X$ and low-precision elements $P_1, \ldots, P_k$
  2: $m \leftarrow \max_i \big(|V_i|\big)$
  3: $shared\_exp \leftarrow \big\lfloor \log_2(m) \big\rfloor - e_{\max}^{\text{elem}}$
  4: $X \leftarrow 2^{shared\_exp}$                       ▷ block scale (a power of two)
  5: **for** $i \leftarrow 1$ **to** $k$ **do**
  6:     $r \leftarrow V_i / X$
  7:     $P_i \leftarrow \textsc{QuantizeToLP}(r)$                ▷ clamp if $|r|$ overflows
  8: **end for**
  9: **return** $(X,\ \{P_i\}_{i=1}^k)$

---

The shared scale in MX formats can therefore be regarded as the largest power-of-two that can represent the maximum within a block, shifted by the exponent of the largest normal value in that type.

### A.1 GEMM simulation settings

We emulate MX (shared–scale) GEMMs using the public PyTorch MX Emulation library Microsoft (2024) and defer to their README for helpful visualizations of where the quantization step happens. For each matrix multiply, the simulation proceeds as follows:

1. **Inputs are quantized to MX.** The high-precision activation $A_{i-1}$ and weight $W_i$ are block-quantized using Algorithm 1 to produce the MX representation.

2. **matmul accumulates in high precision.** The matmul consumes emulated FP8 inputs but performs accumulation in FP32. The matmul output tensor $A_i[M, N]$ is therefore FP32.

3. **High-precision write-back.** The FP32 accumulator result is rounded *once* to $\texttt{bfloat16}$ before the next operation (e.g., bias addition, activation, or the next layer). It is not re-quantized to FP8 at this stage.

Elementwise vector operations (e.g., residual additions and the arithmetic inside layernorm) are executed in $\texttt{bfloat16}$: operands are cast to $\texttt{bfloat16}$ and the operation itself runs in $\texttt{bfloat16}$.

## B Multiplicative Noise

Our synthetic experiments reveal that training instabilities in low-precision settings can arise from both stochastic optimization effects and quantization-induced bias. These failures appear to result from a complex interplay between architectural choices, activation functions, layer normalization, and hyperparameters. One hypothesis, motivated by the growth of the gradient norm in Figure 1, is that lower precision is systematically biasing the gradient. In this section, we examine this hypothesis through a multiplicative noise model and show that it is consistent with the instability patterns seen in low-precision training.

### B.1 Behavior of the Noise

Let

$$\varepsilon_t \equiv \widetilde{g}_t - \bar{g}_t, \tag{4}$$

where $\bar{g}_t$ denotes the exact gradient at time step $t$, and $\widetilde{g}_t$ is its low-precision counterpart. Under a multiplicative noise model, we posit that

$$\widetilde{g}_t = (1 + \boldsymbol{\zeta}_t)\bar{g}_t, \tag{5}$$

where $\zeta_t$ is a (possibly data and parameter-dependent) noise matrix induced by quantization. Although $\zeta_t$ is not directly measurable (and may not even be uniquely defined e.g., due to weight permutations), we can estimate the magnitude of its effect. Specifically, the deviation vector $\varepsilon_t$ satisfies

$$\|\varepsilon_t\|_2 \leq \|\zeta_t\|_{\text{op}}\|\bar{g}_t\|_2, \tag{6}$$

where $\|\cdot\|_{\text{op}}$ denotes the operator norm. In Section B.2, we argue for a heuristic bound that $\|\zeta_t\|_{\text{op}}$ must satisfy through training and how a runaway loss divergence may occur in this model.

To test this model empirically, we replicate the synthetic experiment setup from Section 4. For each configuration, we fix the random seed and weight initialization, then train one model in FP32 to log the exact gradient $\bar{g}_t$ at each step. We then retrain the same model under MXFP8 precision and compute the deviation $\varepsilon_t = \widetilde{g}_t - \bar{g}_t$ at every step. This allows us to extract both the norm ratio $\|\varepsilon_t\|_2/\|\bar{g}_t\|_2$ and the cosine similarity between $\widetilde{g}_t$ and $\bar{g}_t$.

Results are shown in Figure 12. Early in training, the estimate of $\|\zeta_t\|_{\text{op}}$ (as inferred from Equation (6)) gradually decreases. However, as training progresses, the estimate begins to rise. Once $\|\zeta_t\|_{\text{op}} \sim 2$, we observe divergence in the loss. A similar trend is observed in the cosine angle between gradients: it slowly degrades over several thousand steps and eventually flatlines near zero, indicating that the low-precision gradient is no longer aligned with the true descent direction.

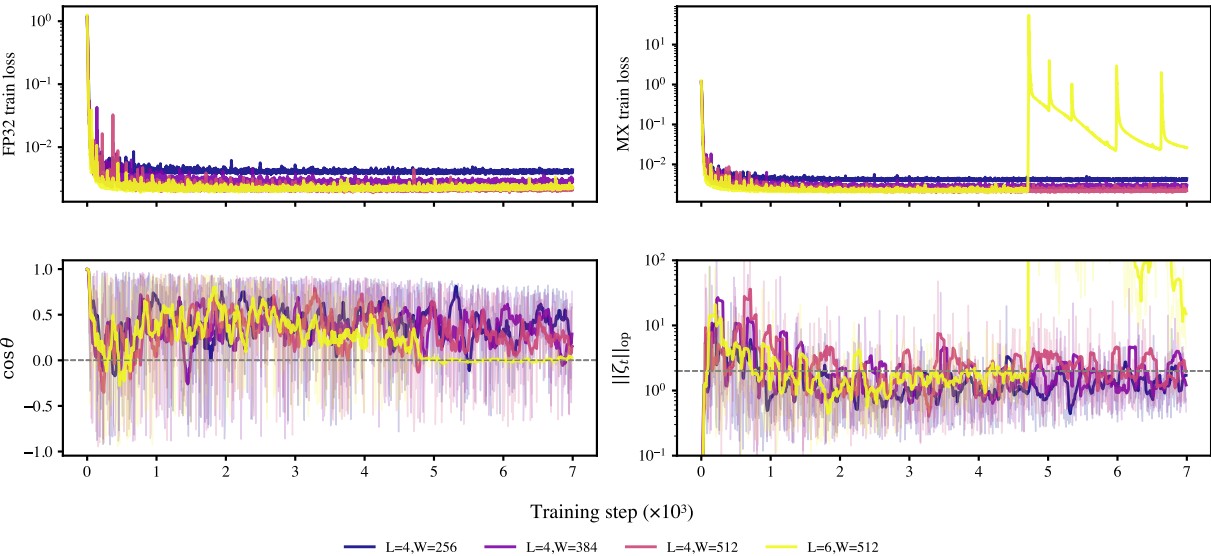

Figure 12: Shows the bound on the operator norm $\|\zeta_t\|_{\text{op}}$ (as inferred from Equation (6)), and the cosine angle between the low precision gradient and high precision gradient. Dashed line in the lower right plot shows when the bound on $\|\zeta_t\|_{\text{op}}$ is equal to 2.

## B.2 A Crude Bound

To understand the behavior of $\|\zeta\|_{\text{op}}$, consider that we have some optimum $w_*$ such that $\nabla_w L(w_*) \approx 0$. Linearizing around the minimum we have

$$\nabla_w L(w_t) = H(w_t - w_*), \tag{7}$$

where $H = \nabla_w^2 L$ is the Hessian. The equation above makes no reference to precision – the only approximation we've made is ignore terms of order $(w_t - w_*)^2$ and higher. Defining $\delta_t \equiv w_t - w_*$, we then have

$$\bar{g}_t = H\delta_t. \tag{8}$$

With some manipulations the GD update rule is[4]

$$\delta_{t+1} = \delta_t - \eta_t(I + \boldsymbol{\zeta}_t)H\delta_t \tag{9}$$

and so

$$\delta_{t+1} = (I - \eta_t H)\delta_t - \eta_t \boldsymbol{\zeta}_t H\delta_t. \tag{10}$$

We can therefore see that there is a driving term proportional to the noise $\boldsymbol{\zeta}_t$; if the noise operator norm is large enough, it can flip a contracting direction into an expanding one. The stability criteria is therefore that the operator $I - \eta_t(1 + \boldsymbol{\zeta}_t)H$ has spectral radius less than one. In terms of the maximum eigenvalue of $H$, $\lambda_{\max}$, this means that a crude bound for stability is

$$|1 - \eta_t \lambda_{\max}| + \eta_t \|\boldsymbol{\zeta}_t\|_{\mathrm{op}} \lambda_{\max} \lesssim 1. \tag{11}$$

Clearly, when the norm of $\boldsymbol{\zeta}_t$ grows, the region of stable $\eta_t \lambda_{\max}$ shrinks. However, from the "edge of stability" viewpoint of Cohen et al. (2021), in the absence of multiplicative noise, $\lambda_{\max}$ is expected to increase until it hovers at or just above $\sim 2/\eta$. Once the multiplicative term $\boldsymbol{\zeta}_t$ is introduced, we may then expect that the stability region defined by Equation (11) contracts. Developing a precise theory for this regime – building on the analysis of Jastrzebski et al. (2020); Damian et al. (2023); Cohen et al. (2021) – is an interesting direction for future work. In the meantime, we bypass an explicit spectral calculation by estimating a lower bound on $\|\boldsymbol{\zeta}_t\|_{\mathrm{op}}$ directly in our synthetic experiments through Equation (6). Empirically, we observe a pattern where the running average of this lower bound first drifts downward, later turns upward (lower right of Figure 12). When it stabilizes around $\approx 2$, training instabilities tend to follow; this observation marks a strong (but not perfect) qualitative correlate of divergence.

## C    Hyperparameter Tuning in our Proxy Model

A key point we aim to distinguish is that there are two classes of instabilities we typically encounter when training models. The first type arises due to incorrect hyperparameter choices. For example, if the size of the steps are large due to, e.g., a large learning rate, this will be visible as a sudden spike(s) in the loss. These types of instabilities are generally recoverable. The second type involves a more serious issue with gradient bias, of the type characterized in Appendix B. In this case, optimization cannot recover since the errors in the gradient computation can compound. In this section, we explain how we tune hyperparameters to avoid the first class of instabilities.

### C.1    Sweeping over learning rates and architectures

**Learning rates**    To illustrate how learning rates can impact stability, we begin by sweeping over learning rates $\eta \in (1 \times 10^{-5}, 5 \times 10^{-5}, 1 \times 10^{-4}, 5 \times 10^{-4}, 1 \times 10^{-3})$ across a range of model depths and widths, in two low precision formats: (1) MXFP8 `E4M3` in the forward pass and MXFP8 `E5M2` in the backward pass[5], and (2) MXFP6 `E4M3` in both forward and backward passes.

Results from this sweep are shown in Figure 13. We observe the following patterns: for sufficiently low learning rates $\eta \lesssim 1 \times 10^{-4}$, all precision formats remain stable. At $\eta = 5 \times 10^{-4}$, differences between FP32 and lower-precision formats begin to emerge: FP32 exhibits two unstable runs, while FP8 shows six. At the highest learning rate ($\eta = 1 \times 10^{-3}$), instabilities are observed across all formats, with larger models failing earlier in training. Interestingly, we find that recovery from an instability is more rapid in FP32, whereas instability in lower-precision formats–particularly FP6–is often more persistent.

We also experimented with a cosine learning rate schedule that starts at $1 \times 10^{-3}$ and decays to $1 \times 10^{-5}$ and found that the effect of the schedule was mainly to suppress instabilities at later training times, though we still observe the same differences between high and low precision if the instability does not happen late in training.

---

[4]Strictly speaking, we are using the stochastic Adam update rule and not GD in our experiments, and so the resulting bound should not be regarded as rigorous nor faithfully represent the LLM setting.

[5]We use this asymmetric format to allow greater dynamic range in the backward pass, following Micikevicius et al. (2022b), and because it exhibited marginally greater stability than using `E4M3` for both passes. Our results are not sensitive to this particular choice of low-precision formats.

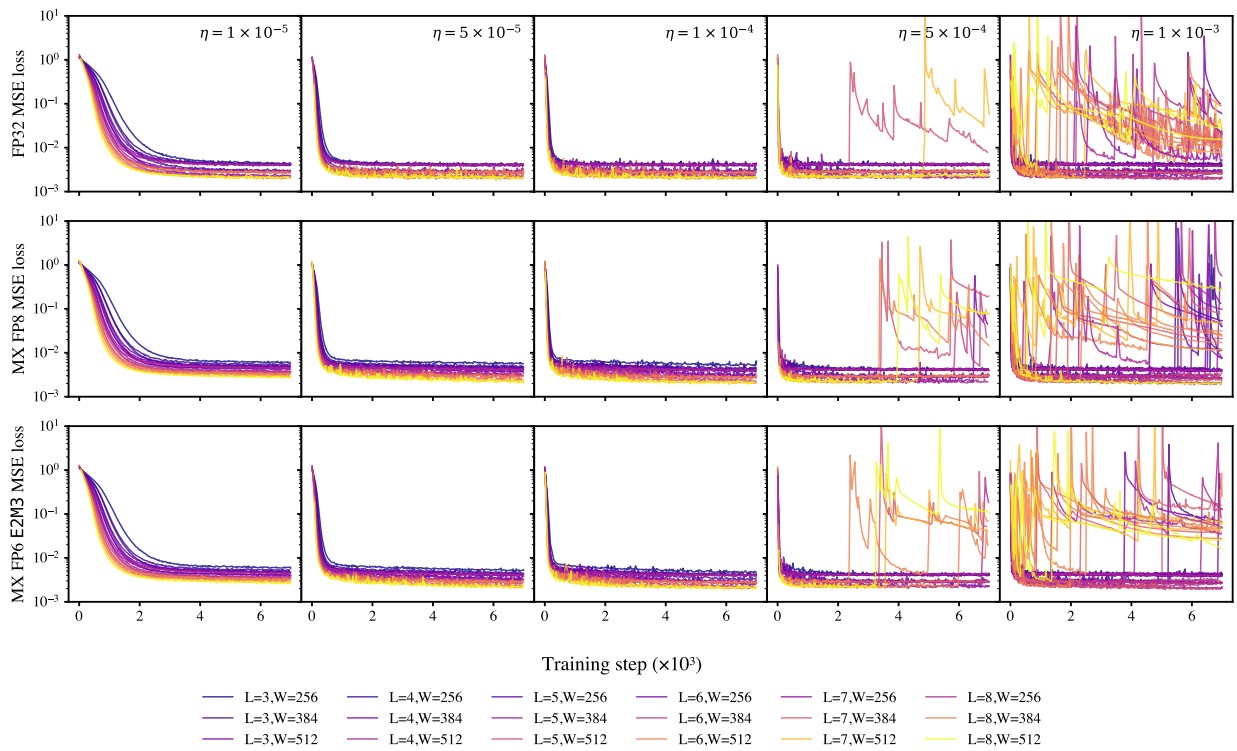

Figure 13: Comparing FP32 with MXFP6 and MXFP8 formats across different choices for the learning rate. Color corresponds to model size, determined by the depth $L$ and $d_{\text{model}} = D$ on the legend.

We find that instability differences between high and low precision seem to occur more frequently in networks of intermediate size, for model dimensions in the range $384 \lesssim d_{\text{model}} \lesssim 768$ and depths $3 \lesssim L \lesssim 6$. Intuitively, this makes sense since these models appear to be large enough to exhibit sensitivity to low-precision effects, yet not large enough where overall stochasticity causes generally unstable training at this learning rate.

**Fixing LR to rule out tuning error** To isolate precision effects, for each $(L, d_{\text{model}})$ we select an LR that yields *no* instabilities in FP32 with GeLU activation and hold it fixed when comparing precisions or performing ablations. While a fully principled approach would use $\mu$P (Yang et al., 2022) to scale LRs with width, in practice, a manual grid search is sufficient due to the small size of the proxy model. We find that there is a range of acceptable learning rates that seem to work well in which high precision runs are stable and low-precision runs are not, for each depth and width. For example, for $3 \lesssim L \lesssim 6$, learning rates in roughly $[2 \times 10^{-4}, 6 \times 10^{-4}]$ are very reliably stable in FP32 yet can be unstable in low precision. As depth/width increase, the stability region for low-precision narrows and requires lower learning rates, even when FP32 remains stable at comparatively larger learning rates.

# D  Differences Between our Proxy Model and LLM

One potential limitation of our proxy model is that it omits certain architectural components of the LLM (most notably self-attention) and that it is trained with mean–squared error (MSE) rather than cross-entropy, reflecting the distributional learning task we study in the synthetic setting.

In this section we ablate both choices. First, we show that the instability we observe already appears without self-attention. Second, we add self-attention to the proxy and find that, perhaps surprisingly, attention can

be stabilizing in some regimes. These results suggest that the primary failure modes we study are not driven by the attention mechanism itself (at least at the scales probed here).

To incorporate attention into our model given in Equation (1), we consider the modifications

$$
\begin{aligned}
A_0 &= x \\
z_k &= A_{k-1} + \text{SelfAttn}(\text{LN}_1(A_{k-1})). \\
A_{k>0} &= z_k + \mathbf{W}_k^{(2)}\phi(\mathbf{W}_k^{(1)}\text{LN}_2(z_k))
\end{aligned}
\tag{12}
$$

That is, the we employ self attention with no mask with pre-attention layernorm. For the attention ablation we treat inputs as sequences (shape $(B, S, d_{\text{model}})$) to enable "token–token" interactions although this is a synthetic sequence dimension introduced solely for the ablation. Given a fixed compute budget, increasing $S$ typically requires reducing the batch size $B$. In general, we do not find that including attention causes additional instability, suggesting that the primary failure mode is not caused by attention itself. An example training run with this ablation is shown in Figure 14. In this instance, adding self-attention actually improves training stability in the low-precision setting.

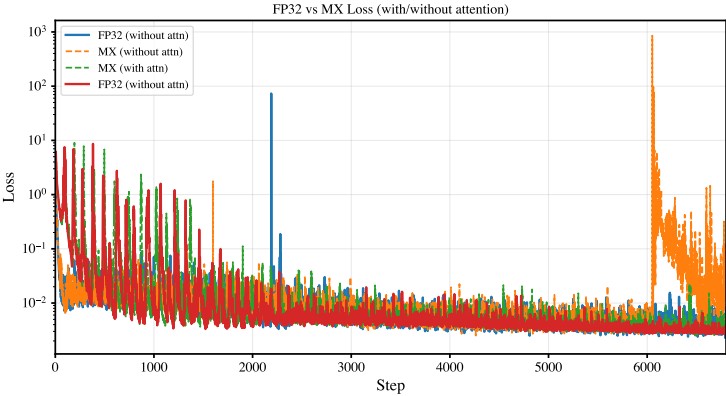

Figure 14: Shows an example synthetic training run where we ablated over self-attention in the proxy model, in both FP32 and MX FP8 E4M3. The orange run (low precision without attention) is more prone to instability across training runs.

Next, we evaluate the impact of using an MSE loss in our proxy model. In Figure 15, we evaluate stability in low precision when using MSE loss versus a KL loss on the softmax of the logits (with temperature 1). Both runs eventually diverge, although optimization seems to recover more quickly in the KL setting.

### D.1  Where the Proxy Model and Language Model Training Runs Differ

For completeness, we record the disagreements between our proxy model and the language model training runs.

The proxy model in Equation (1) is a stripped-down stand-in for an LLM: its purpose is to act as a probe of how MX block scaling interacts with the components common to both settings (residual stream, layer normalization with affine parameters, point-wise nonlinearity, two-matrix MLP blocks). Because this is a probe and not a faithful model of an LLM, we treat agreement between the proxy and the LLM as supporting evidence for a shared mechanism rather than a causally predictive model. In this section we make this stance precise: we enumerate which architectural ingredients the proxy retains and which it omits and summarize where the proxy and the LLM agree and disagree quantitatively, so the reader can judge how much weight to place on each proxy-derived conclusion.

**Architectural and training-setup coverage.**  Table 3 summarises which features of the LM stack we retain in the proxy. The proxy captures the components that the LN-overflow mechanism depends on (a

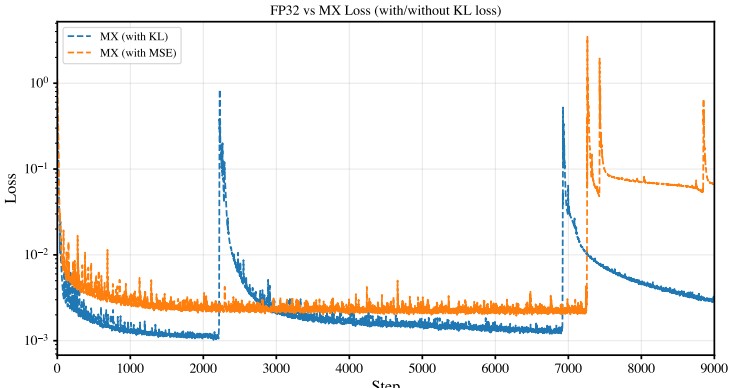

Figure 15: Shows an example synthetic training run where we used a KL loss on softmax logits compared to an MSE loss. Both runs eventually diverge (while full precision is stable), although the instability is less recoverable in the MSE case.

residual stream that accumulates quantization bias step-by-step, LN with affine parameters $\gamma, \beta$ in every block, MX quantization in both forward and backward passes, and the same nonlinearity family as the LM sweep), while omitting components we did not find to be necessary for reproducing the divergence (self-attention, embeddings, multiple LN flavours such as QK-norm and FFN-norm, positional encoding, cross-entropy on a real token distribution).

| Feature | Proxy | LM | Notes |
|---|---|---|---|
| Residual stream + pre-norm structure | ✓ | ✓ | Both accumulate bias through the residual path. |
| LN affine parameters $(\gamma, \beta)$ | ✓ | ✓ | The main component of gradient bias (other being activations). |
| MX quantization, fwd *and* bwd | ✓ | ✓ | Same MX PyTorch emulation library on both sides. |
| GeLU / SwiGLU / ReLU sweep | ✓ | ✗ | LM sweep is GeLU only, proxy sweeps all. |
| Adam optimizer | ✓ | ✓ | |
| Same-seed FP32 vs. MX controls | ✓ | ✗ | Only the proxy permits exact-state retraining. |
| Self-attention (with mask, softmax, QK-norm) | ✗ | ✓ | We ablate unmasked self-attention in App. D but not causally-masked. |
| Multiple LN flavours (FFN-LN, attn-LN, QK-LN) | ✗ | ✓ | Proxy has a single LN per block. |
| Token embeddings / unembeddings | ✗ | ✓ | |
| Real text token distribution | ✗ | ✓ | Proxy inputs are i.i.d. Gaussian. |
| Channel-wise activation outliers ($\sim 100\times$ median) | ✗ | ✓ | Gaussian inputs do not reproduce the outlier tail observed in real LMs. |
| Cross-entropy loss with softmax saturation | ✗ | ✓ | We test a KL variant in App. D; same qualitative picture. |
| Rotary / positional encoding | ✗ | ✓ | |

Table 3: Coverage of LM-stack features in the proxy model.

**Matched configurations we checked** We did not run a fully paired sweep across all $\sim$1000 LM configurations and a matched proxy configuration for each, as the proxy and the LM live at different parameter scales by design. What we did check was that a small set of paired experiments where a specific intervention or ablation was applied identically (in spirit) to both settings, and we can report the agreement on these. Table 4 lists them.

| Intervention / setting | Proxy outcome | LM outcome | Agreement |
|---|---|---|---|
| MXFP8 E4M3-E4M3, GeLU, baseline (no mitigation) | Unstable (e.g., Fig. 4) | Unstable in larger runs (Fig. 22) | Agree |
| MXFP8 E5M2-E5M2, GeLU, baseline | Unstable for some depths/widths (Fig. 13) | Unstable for larger runs (Fig. 1b) | Agree |
| Disable LN affine quantization | Stabilizes/delays (Fig. 4) | Delays at LR=2e-4 (Fig. 20) | Agree |
| Activations in bfloat16, weights MXFP8 | Stabilizes if intervened early enough (Fig. 4) | Recovers near-bfloat16 scaling (Tab. 1) | Agree |
| Quantize forward pass only | Stabilizes (Fig. 4) | Recovers near-bfloat16 scaling (Tab. 1) | Agree |
| Bump shared exponent by one | No mitigation (Fig. 4) | Not tested in LM | Not tested |
| SwiGLU vs GeLU | SwiGLU more unstable (Fig. 2a) | Not tested in LM (GeLU only) | Not tested |

Table 4: Paired interventions/settings we checked across both the proxy and the LM, with the agreement from each.

## E    Additional Synthetic Sweeps

In this section, we present additional synthetic experiments to further examine the sources and mitigation of low-precision instabilities.

Figure 16 summarizes the frequency of instability spikes across our depth-width sweep at a fixed learning rate of $\eta = 5 \times 10^{-4}$. The MX-mix format refers to the asymmetric configuration using `MXFP8 E4M3` in the forward pass and `E5M2` in the backward pass. Spikes were determined by the heuristic criteria that the loss at time step $t$ had to be a factor of 100 lager than the loss at time step $t-1$; this gives a rough lower bound on the number of spikes.

Figure 17 compares the impact of optimizer choice, focusing on SGD with momentum, and vanilla SGD (momentum = 0). These experiments used a slightly higher learning rate of $\eta = 1 \times 10^{-2}$ to exaggerate differences. Compared with Figure 13, we observe that SGD variants are more stable than Adam, perhaps due to Adam's use of second-moment accumulation, which may amplify quantization-induced bias in low-precision regimes.

Figure 18 evaluates the effect of different weight initialization schemes. We compare standard Pytorch initialization, typically taken to be a Kaiming uniform distribution between $[-1/\sqrt{\text{fan in}}, 1/\sqrt{\text{fan in}}]$, against a variant using lower gain (`gain = 0.5`) under the Xavier normal distribution. Reducing the variance of initial weights appears to improve loss spikes.

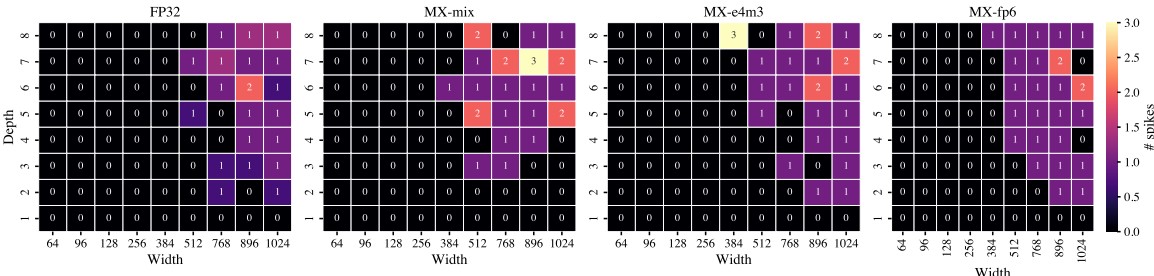

Figure 16: Instability spikes measured in training, for different model depths and widths.

Another sweep we tried was replacing LayerNorm with RMSNorm to see if we still observed the overflow effect (Figure 19). We used a proxy configuration with four layers, model dimension of 512, GeLU activation, MXFP8 E4M3 weights and activations (forward and backward), batch size 2048, peak LR 6e-4, but swapping

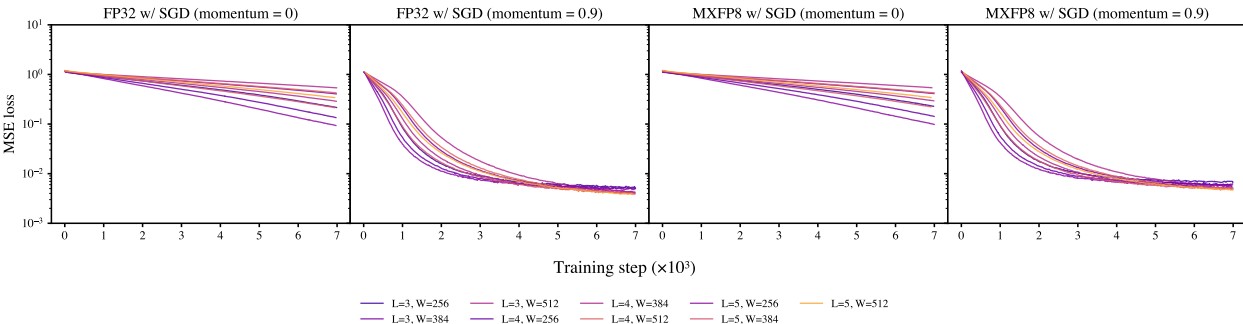

Figure 17: SGD with and without momentum; a larger learning rate was used $\eta = 1 \times 10^{-2}$.

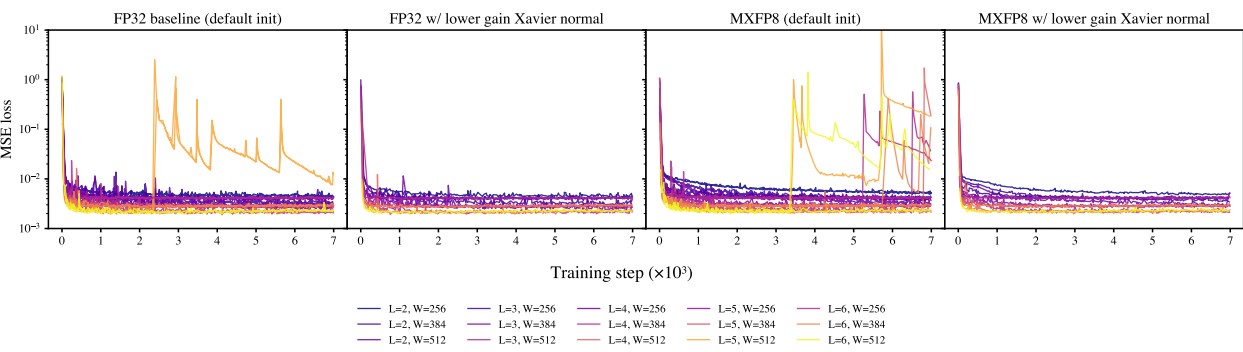

Figure 18: Baseline versus using a lower gain Xavier normal weight initialization.

LayerNorm for RMSNorm (learnable $\gamma$) as the only change. The matched LayerNorm baseline used the same hyperparameters. We observed that, for example, RMSNorm's layer-0 $\gamma$ saturates just as fast than the corresponding LayerNorm $\gamma$ and actually remains fully saturated for the rest of training, similar to LayerNorm. We also observe similar degrees of activation overflow between the two.

## F  LayerNorm Ablations on LM Setting

Here, we show results when we disable layernorm affine weights in the language model setting. The result is shown in Figure 20. In general, with all else being equal, disabling layernorm weights does stabilize training significantly compared to the same run with affine weights. However, eventually the run does become unstable, potentially due to overflow effects in critical activations in the network. For a lower learning rate, disabling affine weights almost completely stabilizes training compared to enabling them i.e. it enlarges the stability window.

## G  Activation Underflow in LM

In Section 5, we pointed out that there is an overflow effect induced by clustering of LayerNorm weights. In this section, we examine a related but different effect of *underflow*.

Following the MX recipe in Algorithm 1 and the GEMM-simulation conventions of Section A.1, for a block $\mathbf{V} \in \mathbb{R}^{32}$ with absolute maximum $m = \max_i |V_i|$, the shared scale is

$$X = 2^{\lfloor \log_2 m \rfloor - e_{\max}^{\text{elem}}}, \tag{13}$$

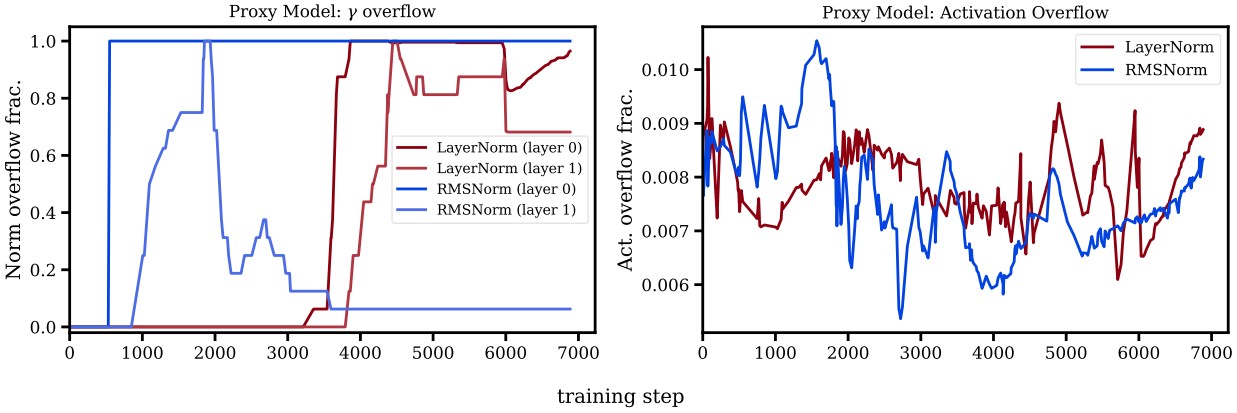

Figure 19: Proxy model training run tracking RMSNorm overflow vs. LayerNorm overflow.

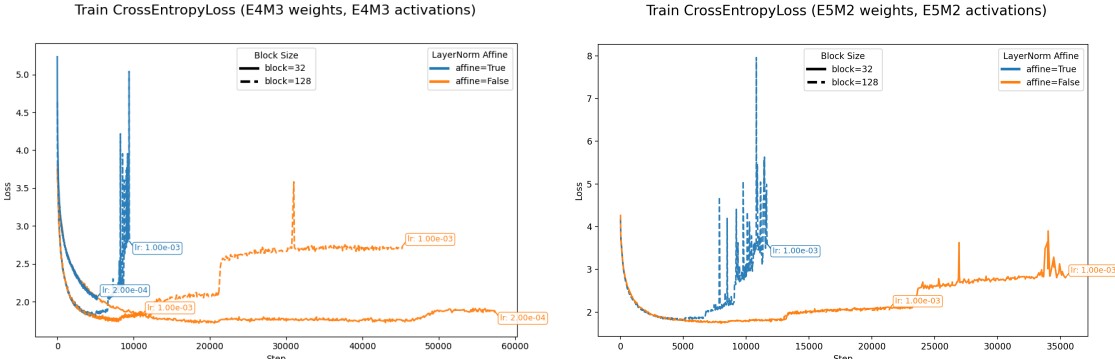

(a) Train loss for weights and activations in MXFP8 E4M3-E4M3 format.

(b) Train loss for weights and activations in MXFP8 E5M2-E5M2 format.

Figure 20: Shows that at the same learning rate (2e-4), turning off affine parameters stablizes the training, while learning rate 1e-3 again makes the training unstable.

so that $V_i/X$ lies, by construction, no higher than the largest representable normal in the chosen element format. For `E4M3`, $e_{\max}^{\text{elem}} = 8$ and the largest representable normal is 448, the smallest representable subnormal is $2^{-9}$, and the smallest value that does *not* round to zero under round-to-nearest-even is $2^{-10}$ (half of the smallest subnormal). Define the outlier ratio for the block as $\rho \equiv m/|V_i|$. Then:

- $V_i$ rounds to a *subnormal* in `E4M3` iff $|V_i/X| < 2^{-6}$ (the smallest normal), which after substituting equation 13 and using $\lfloor \log_2 m \rfloor \leq \log_2 m$ gives the sufficient condition

$$\rho \; > \; 2^{e_{\max}^{\text{elem}}+6} \; = \; 2^{14} \; \approx \; 1.64 \times 10^4. \tag{14}$$

- $V_i$ rounds to *zero* iff $|V_i/X| < 2^{-10}$, giving

$$\rho \; > \; 2^{e_{\max}^{\text{elem}}+10} \; = \; 2^{18} \; \approx \; 2.62 \times 10^5. \tag{15}$$

That is, in MXFP8 `E4M3` a non-outlier value can only be flushed to zero if it is more than five orders of magnitude smaller than the largest value in its 32-element block.

To see how prominent this effect is empirically, we ran the following experiment: we trained a run at both MXFP8 `E4M3` precision and a matched bfloat16 baseline. Both use an OLMo model with 8 layers,

$d_{\text{model}} = 1024$, 16 heads, and LayerNorm ($\sim$100M non-embedding parameters, same architecture family as our main sweep, with other configurations kept the same as in Section H). Both train for a total of $\sim$2.1B Fineweb-Edu tokens (i.e. Chinchilla-optimal).

Every 100 steps we logged the post-attention-LN and post-FF-LN activations for every layer. Each tensor was then re-quantized offline with the MX library (`E4M3`, block size 32) and the per-block underflow fraction, relative MSE, and outlier ratio $\max|x|/\text{median}\,|x|$ were computed. Results are in Figure 21.

Outlier severity, averaged over the eight transformer layers, grows monotonically from $\sim$5$\times$ at step 100 to $\sim$18$\times$ at step 4,000 in *both* the MXFP8 and bfloat16 runs (Figure 21, left). The trajectories are consistent with each other at our compute scale, indicating that the outlier formation process is driven by the training dynamics rather than by the choice of precision format, as both precision formats seem to show roughly similar outlier ratios.

The fraction of activation values that quantize to zero is $\sim$3–6$\times10^{-5}$ across training in both runs (Figure 21, right). True underflow seems to be rare, compared to the $\sim$0.5–1% layernorm/activation *overflow* fractions we report for the same model in Figure 3.

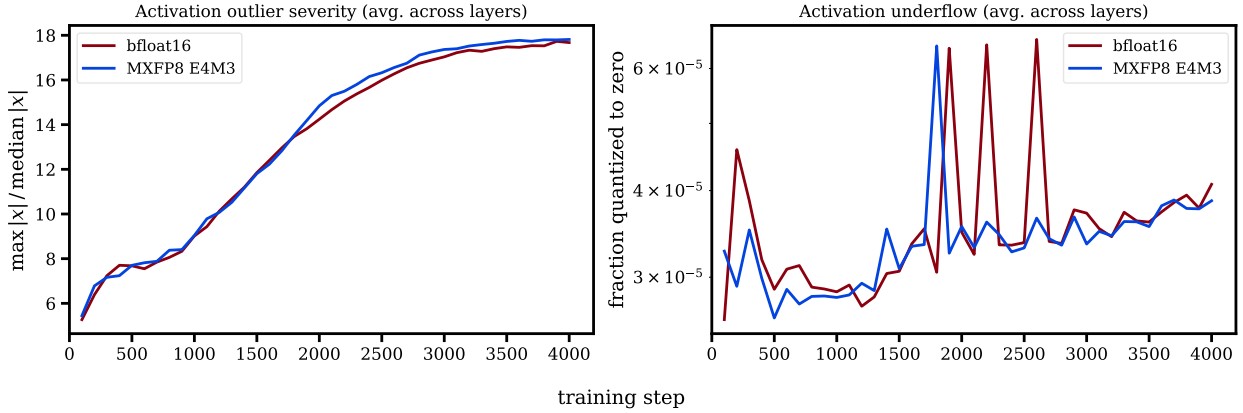

Figure 21: LLM experiment tracking outlier severity and activation underflow due to outliers.

## H    Details of LM Training

All models are trained on the Fineweb-Edu dataset (Penedo et al., 2024) using the Olmo codebase (Groeneveld et al., 2024), with the longest runs trained on 35B tokens and the shortest runs corresponding to models trained on 301M tokens. Models were trained with a learning rate schedule with a linear warmup starting at 2e-5 increasing to 2e-4, followed by cosine decay back to 2e-5 (Porian et al., 2025). Training runs that involved using MX precision formats were done performed using MX Pytorch Emulation Library (Microsoft, 2024).

**A comment on the learning rates:**   To pick this LR, we ran a coarse sweep on the bf16 baseline over $\eta \in \{1 \times 10^{-5}, 2 \times 10^{-4}, 1 \times 10^{-3}\}$ at several model sizes in our sweep. At $\eta = 1 \times 10^{-3}$ even bf16 was ocassionally unstable across for larger ($\sim$ 1B parameter) models, though this is not due to the same instability as the low precision formats and more of an artifact of a poorly tuned LR (in particular, optimization is typically able to recover from such instabilities). At $\eta = 2 \times 10^{-4}$ every bf16 run in the sweep converged but several MX runs diverged. Thus, this is the LR we adopted, and it is the value reflected in Table 5. Smaller learning rates ($\eta \leq 1 \times 10^{-5}$) tend to be stable across formats as they trivially make the parameter updates small, but this learning rate is too low to use in practice for the model sizes we tested.

| Parameter | Value |
| --- | --- |
| $n$ | 6–24 for small models |
| Number of heads | $n$ |
| Head dimension | 64 |
| MLP hidden multiplier | 4 |
| Depth | $n$ |
| Context length | 512 |
| Activation | GeLU |
| Positional encoding | RoPE |
| Biases | False |
| Normalization | PyTorch layernorm |
| QK normalization | True |
| Tokenizer | Llama2 |

Table 5: Model parameters used for training.

## I  Validation Losses in Language Models with Mitigations

Table 6 and continued in 7 shows validation losses for all models with mitigations applied (quantization only in the forward pass, or activations in high precision), trained using our at different FLOP budgets relative to bfloat16 baseline.

| formats D/N | | bfloat16 bfloat16 | E4M3 bfloat16 | E5M2 bfloat16 | E4M3 E4M3 | E5M2 E5M2 |
|---|---|---|---|---|---|---|
| **87.35** | | 1.1522 | -0.027 | -0.027 | -0.027 | -0.012 |
| **46.99** | | 1.1084 | 0.002 | 0.007 | 0.007 | 0.012 |
| **26.897** | | 1.1011 | 0.004 | 0.001 | 0.004 | 0.009 |
| **16.06** | 2e+17 | 1.0956 | -0.001 | 0.014 | 0.004 | 0.009 |
| **9.92** | | 1.0971 | 0.003 | 0.003 | 0.008 | 0.013 |
| **6.30** | | 1.0950 | 0.0 | -0.005 | -0.01 | 0.01 |
| **4.10** | | 1.1042 | 0.001 | -0.006 | 0.006 | 0.006 |
| **2.73** | | 1.1255 | -0.001 | -0.004 | 0.004 | 0.019 |
| **191.02** | | 1.030 | 0.005 | 0.0 | 0.010 | 0.01 |
| **102.78** | | 1.0464 | -0.016 | 0.036 | -0.011 | -0.021 |
| **58.81** | | 0.9898 | 0.005 | 0.005 | 0.005 | 0.015 |
| **35.14** | | 0.9806 | -0.001 | 0.004 | 0.004 | 0.009 |
| **21.70** | 4.37e+17 | 0.9765 | 0.003 | 0.003 | 0.003 | 0.013 |
| **13.78** | | 0.9717 | 0.003 | 0.003 | 0.003 | 0.008 |
| **8.97** | | 0.9732 | 0.002 | 0.002 | 0.002 | 0.012 |
| **5.97** | | 2.3174 | 0.303 | 0.843 | 2.763 | 1.237 |
| **4.05** | | 0.9839 | 0.001 | 0.006 | 0.006 | 0.006 |
| **2.80** | | 0.9949 | 0.0 | 0.0 | 0.0 | 0.005 |
| **128.62** | | 0.9198 | 0.0 | 0.0 | 0.005 | 0.015 |
| **76.84** | | 0.9052 | 0.0 | 0.005 | 0.005 | 0.015 |
| **47.46** | | 0.8969 | 0.002 | 0.003 | 0.003 | 0.008 |
| **30.14** | | 0.8894 | 0.001 | 0.001 | 0.006 | 0.011 |
| **19.62** | | 0.8846 | 0.0 | 0.005 | 0.005 | 0.01 |
| **13.05** | 9.56e+17 | 0.8879 | 0.002 | 0.002 | 0.002 | 0.012 |
| **8.86** | | 0.8849 | 0.0 | 0.005 | 0.005 | 0.005 |
| **6.13** | | 0.8882 | 0.002 | 0.002 | 0.002 | 0.007 |
| **4.31** | | 0.8933 | 0.002 | 0.002 | 0.002 | 0.007 |
| **3.08** | | 0.8961 | 0.004 | 0.004 | 0.004 | 0.009 |
| **2.24** | | 0.9059 | -0.001 | 0.004 | 0.064 | 0.004 |
| **168.03** | | 0.8546 | 0.0 | 0.005 | 0.005 | 0.015 |
| **103.78** | | 0.8430 | 0.002 | 0.002 | 0.187 | 0.012 |
| **65.91** | | 0.8335 | 0.001 | 0.001 | 0.001 | 0.011 |
| **42.896** | | 0.8258 | -0.001 | 0.004 | 0.004 | 0.009 |
| **28.54** | | 0.8242 | 0.001 | 0.001 | 0.001 | 0.011 |
| **19.37** | 2.09e+18 | 0.8200 | 0.0 | 0.0 | 0.0 | 0.005 |
| **13.399** | | 0.8197 | 0.0 | 0.0 | 0.0 | 0.005 |
| **9.428** | | 0.8187 | 0.001 | 0.001 | 0.001 | 0.006 |
| **6.74** | | 0.8192 | 0.001 | 0.001 | 0.001 | 0.006 |
| **4.89** | | 0.8215 | 0.003 | 0.006 | 0.003 | 0.003 |
| **2.02** | | 0.8327 | 0.002 | 0.002 | 0.002 | 0.002 |

Table 6: Validation loss table, with separate columns for various weight and activation precisions. For the last 2 columns, we quantize only the forward pass. The second column indicates the total FLOP count used for those values of tokens-to-parameter ratios ($D/N$). Values are shown as differences with respect to bfloat16 baseline (lower is better).

| formats D/N | bfloat16 bfloat16 | E4M3 bfloat16 | E5M2 bfloat16 | E4M3 E4M3 | E5M2 E5M2 |
|---|---|---|---|---|---|
| **144.14** | 0.794 | 0.001 | 0.006 | 0.006 | 0.011 |
| **93.81** | 0.784 | 0.001 | 0.001 | 0.001 | 0.011 |
| **62.41** | 0.780 | 0.0 | 0.005 | 0.005 | 0.01 |
| **42.37** | 0.774 | 0.001 | 0.001 | 0.001 | 0.006 |
| **29.30** | 0.772 | -0.002 | 0.003 | 0.003 | 0.003 |
| **14.74** | 0.767 | -0.002 | 0.003 | 0.003 | 0.003 |
| **10.70** | 0.766 | -0.001 | 0.004 | -0.001 | 0.004 |
| **7.87** | 0.766 | -0.001 | 0.004 | -0.001 | 0.004 |
| **4.42** | 0.769 | 0.001 | 0.001 | 0.001 | 0.006 |
| **3.37** | 0.772 | -0.002 | 0.003 | 0.003 | 0.003 |
| **2.60** | 0.775 | 0.0 | 0.0 | 0.0 | 0.005 |
| **2.02** | 0.779 | 0.001 | 0.001 | 0.001 | 0.001 |
| **136.47458** | 0.748 | 0.002 | 0.002 | 0.002 | 0.002 |
| **92.646** | 0.741 | -0.001 | 0.004 | 0.004 | 0.009 |
| **64.075** | 0.736 | -0.001 | 0.004 | 0.004 | 0.009 |
| **45.084** | 0.731 | -0.001 | 0.004 | 0.004 | 0.009 |
| **32.233** | 0.728 | 0.002 | 0.002 | 0.002 | 0.007 |
| **23.391** | 0.725 | 0.0 | 0.005 | 0.0 | 0.005 |
| **17.210** | 0.724 | 0.001 | 0.001 | 0.001 | 0.006 |
| **12.826** | 0.724 | 0.001 | 0.001 | 0.001 | 0.311 |
| **9.674** | 0.723 | 0.002 | 0.002 | 0.002 | 0.002 |
| **7.38** | 0.723 | 0.002 | 0.002 | 0.002 | 0.077 |
| **4.43** | 0.727 | -0.002 | 0.003 | 0.003 | 0.003 |
| **2.75** | 0.732 | -0.002 | 0.023 | 0.003 | 0.003 |

The FLOP count column: **4.57e+18** for the first block and **1e+19** for the second block.

Table 7: MXFP8 of the validation loss table, with separate rows for Weight and Activation precisions. For the last 2 columns, we quantize only the forward pass. The second column indicates the FLOP count used.

## J  Additional Unstable LM Sweeps

In Figure 23 and Figure 24 we show some other examples of weight/activation MX precision combinations we found to be unstable. In general, we were not able to find any stable combinations of weights and activations in lower precision across the formats we tested. In Figure 22 we also show a pretraining training run on the StarCoder (Li et al., 2023) dataset, which is comprised of entirely code, as a data point that these divergences are not dataset-dependent.

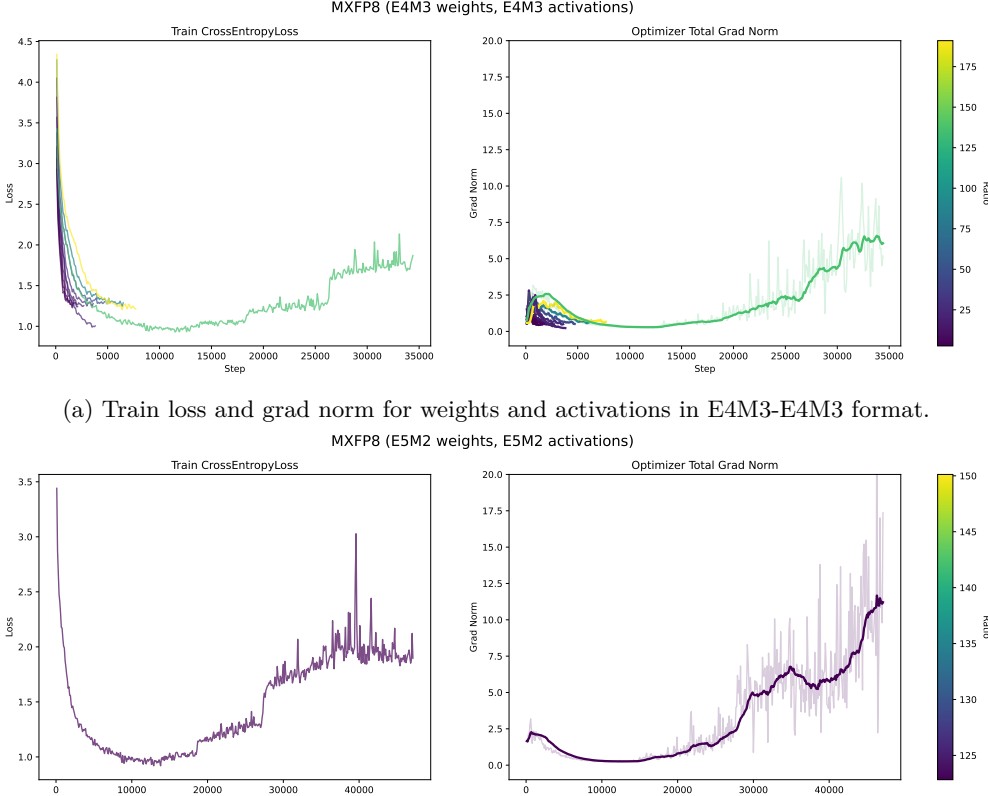

(a) Train loss and grad norm for weights and activations in E4M3-E4M3 format.

(b) Train loss and grad norm for weights and activations in E5M2-E5M2 format.

Figure 22: Shows OLMo training runs (top) on StarCoder. The low precision computations are done in both forward and backward steps, on both weights and activations. Color bar on the right shows the token-to-parameter ratio.

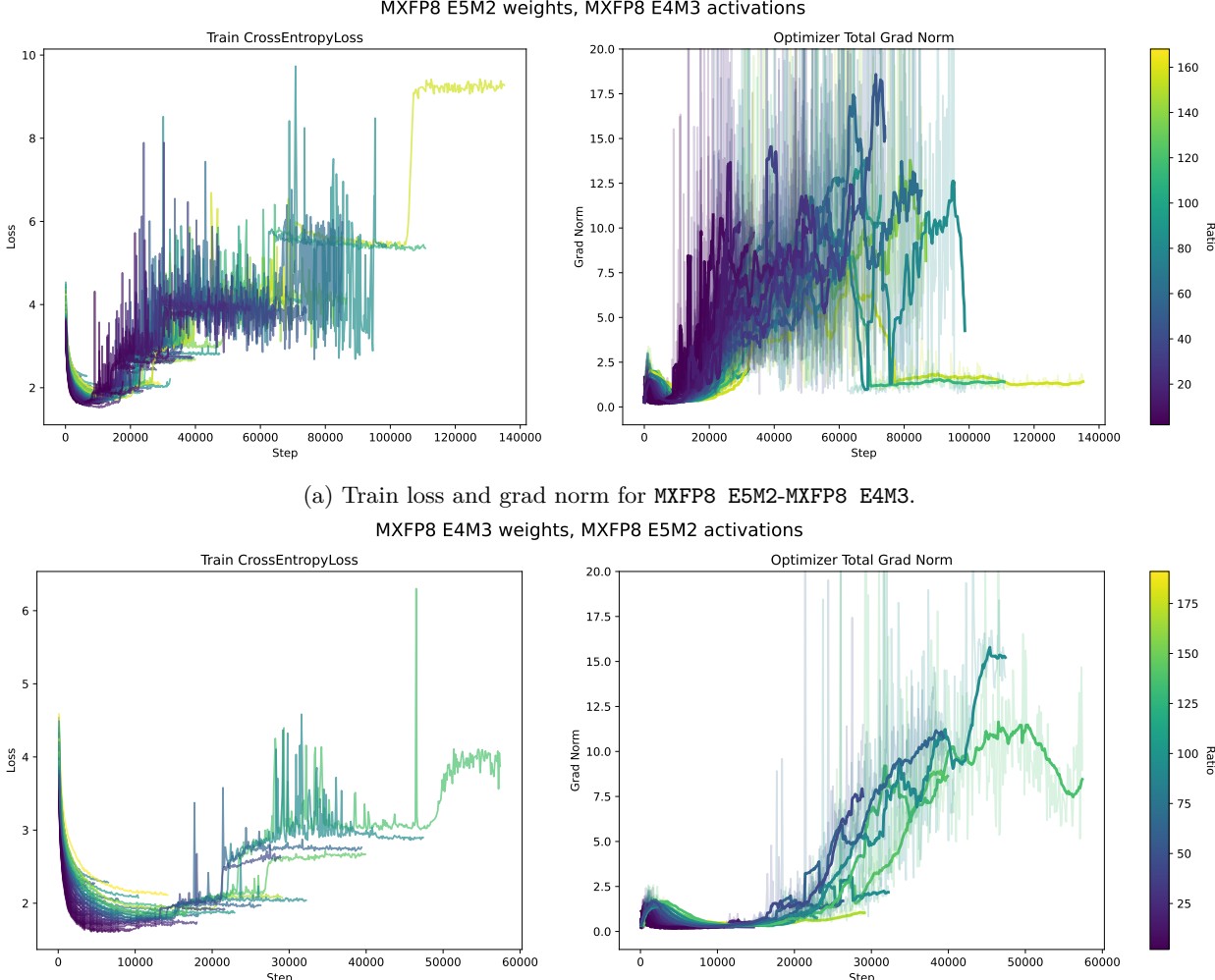

(a) Train loss and grad norm for `MXFP8 E5M2`-`MXFP8 E4M3`.

(b) Train loss and grad norm for `MXFP8 E4M3`-`MXFP E5M2`.

Figure 23: **Unstable** MXFP8 combinations of precision formats of weights and activations.

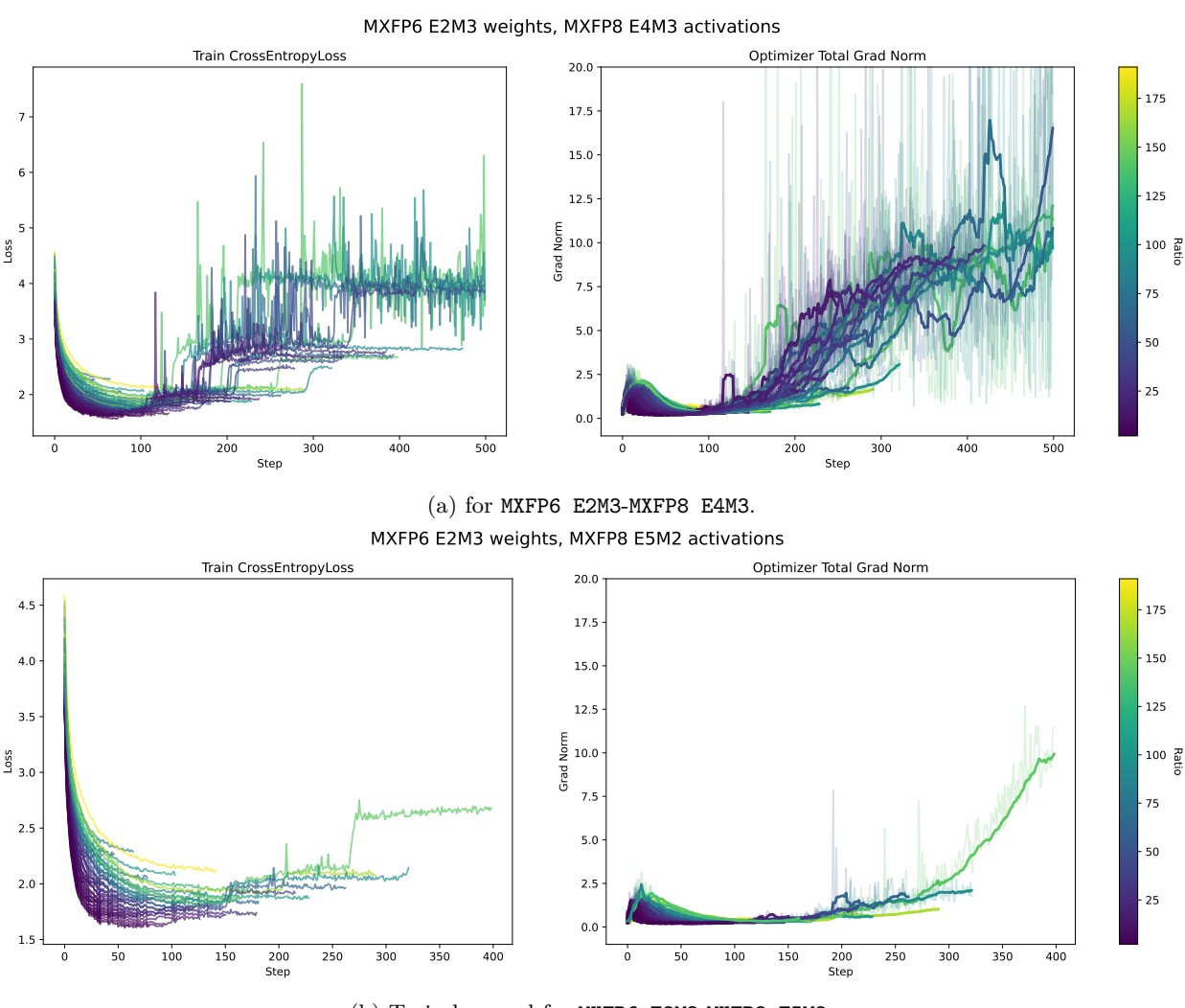

(a) for `MXFP6 E2M3-MXFP8 E4M3`.

(b) Train loss and for `MXFP6 E2M3-MXFP8 E5M2`.

Figure 24: **Unstable** combinations of precision formats of weights and activations for MXFP6 weights.

