# OpenReview forum: "A Mechanistic Analysis of Low-Precision Instabilities in Microscaling Formats"
_TMLR — Accepted by TMLR_

### Review · Reviewer_sSV3 · 2026-04-27

**Summary Of Contributions:**

This paper studies training instabilities under Microscaling (MX) block-scaled formats (MXFP8, MXFP6) for LLM pretraining. The authors train roughly 1000 OLMo models across compute budgets from 2×10^17 to 4.8×10^19 FLOPs, and consistently observe sharp, irrecoverable loss spikes when using MX formats, especially at larger scales. To pinpoint the cause, they construct a residual MLP proxy model in a student-teacher setup and identify a specific failure mode: layernorm affine weights become tightly clustered over the course of training, so that after division by the MX block scale, nearly all values land in the overflow region and get clamped to the largest representable number. This introduces systematic gradient bias that eventually triggers divergence. The authors then test several mitigations (exempting layernorm from quantization, keeping activations in bfloat16, forward-only quantization) and show that hybrid precision configurations can largely recover bfloat16-level performance on the LLM side.

Strengths:

- The sheer scale of LLM experiments (~1000 runs, multiple formats, two datasets) gives strong empirical grounding. The instabilities are clearly not hyperparameter accidents.
- The layernorm overflow mechanism (Section 5.1, Eq. 2) is a clean and concrete finding. The observation that block scaling interacts poorly with tightly-clustered distributions is broadly useful.
- The intervention experiment (Figure 4), where precision is switched mid-run while everything else stays frozen, is a nice way to tease apart causality.


Weaknesses:

- The proxy model differs from real LLMs in ways that are directly relevant to the failure modes being studied (see below).
- The theoretical analysis in Appendix B is acknowledged by the authors as non-rigorous and remains mostly qualitative.
- Mitigations are verified at relatively modest scale compared to where the instabilities are most severe.

**Audience:**

Yes

**Audience Explanation:**

MX format training is practically relevant given Blackwell hardware, and the layernorm overflow mechanism is actionable. The mitigation strategies and scaling law fits (Section 6) are useful for practitioners even independent of the theoretical framing.

**Claims And Evidence:**

Yes

**Claims Explanation:**

The core concern is the proxy model's representativeness. A shallow residual MLP on Gaussian inputs with MSE loss is a reasonable controlled setup, but it omits features that directly bear on the failure modes being studied:

(1) **Activation outliers.** Real LLMs develop channel-wise activation outliers orders of magnitude larger than the median. These outliers dominate the MX block-max, forcing other values in the block to lose precision or underflow after scale division. This is the flip side of the layernorm overflow story: instead of all values being too close (causing overflow), one extreme value pushes everything else too small. A Gaussian-input MLP will not produce this pattern. The paper mentions ~0.5-1% activation overflow (Figure 3 right) but does not discuss outlier-induced underflow at all. Which effect matters more at scale is an open question.

(2) **Self-attention.** The single ablation in Appendix D (unmasked attention on synthetic data) is too limited to rule out attention's role. Real attention involves causal masking, softmax saturation, and QK norm growth, all contributing to outlier formation.

(3) **Mitigation scale.** Table 1 goes up to ~0.5B params. Since instabilities worsen with scale, verifying mitigations only at this range is insufficient. This should be explicitly flagged.

(4) **Appendix B.** The noise model assumes linearized GD while experiments use Adam. The ||ζ_t||_op ≈ 2 divergence threshold is only established on the proxy.

The layernorm overflow finding is real and useful. But the paper claims to explain *the* mechanism behind MX instabilities in LLMs, when it more precisely establishes *a* mechanism in a simplified proxy, plus evidence that mitigating it helps. This gap matters.

**Requested Changes:**

1. The activation outlier issue needs substantive discussion. In real LLMs, outliers can be 100x the median activation in the same layer. How does this interact with MX block scaling? Does the block-max being dominated by an outlier cause underflow for other elements, and is this complementary to or potentially more important than the layernorm overflow? At minimum, compare activation distribution statistics (max/mean ratio, channel-wise variance) between the proxy model and OLMo to show how different they actually are.

2. Either verify mitigations at ≥1B parameter scale, or explicitly discuss why the current verification range may be insufficient and what failure modes might emerge at larger scale.

3. Appendix D is titled "Differences Between our Proxy Model and LLM" but only covers attention and loss function ablations. Either broaden the discussion to cover the more fundamental structural gaps (outliers, embedding layers, vocabulary, real data distributions) or adjust the title.

---

> ### Author Response · Authors · 2026-05-25
> **Addressing Reviewer sSV3 Feedback**
>
> We thank the reviewer for carefully reading our work and for their valuable feedback.  We are especially glad to see they highlight the scale of our experiments and the practical relevance of our work for Blackwell hardware.
>
> We address the stated weaknesses/requested changes below:
>
> # Weaknesses / Requested Changes
>
> 1. > The activation outlier issue needs substantive discussion. In real LLMs, outliers can be 100x the median activation in the same layer. How does this interact with MX block scaling? Does the block-max being dominated by an outlier cause underflow for other elements, and is this complementary to or potentially more important than the layernorm overflow? At minimum, compare activation distribution statistics (max/mean ratio, channel-wise variance) between the proxy model and OLMo to show how different they actually are.
>
> We thank the reviewer for raising an important issue regarding underflow, as outlier effects can indeed cause underflow in real models.  To address this, we have added a new Appendix G ("Activation Underflow in LM") in which we discuss the conditions under which we might expect underflow effects to become visible in the block-scaling precision format.
>
> We also ran an LLM experiment to track the underflow effect empirically.  In particular, we trained two runs: one at MXFP8 `E4M3` precision and a matched `bfloat16` baseline, both with a 100M-parameter model configuration on $\sim$2.1B Fineweb-Edu tokens (i.e. Chinchilla-optimal).  We observe two effects: (1) outlier severity tends to grow during training and is roughly the same between MXFP8 and bf16, and (2) the fraction of activation values that quantize to zero is $\sim$3-6$\times 10^{-5}$ across training in both runs.  This is substantially below the $\sim$0.5-1\% layernorm/activation overflow fractions.  This suggests that the overflow effect is the dominant one for MX formats.  Still, we cannot rule out that additional or stronger underflow effects emerge at larger scale, and the underflow phenomenon should be studied more carefully.
>
> 1. > Either verify mitigations at ≥1B parameter scale, or explicitly discuss why the current verification range may be insufficient and what failure modes might emerge at larger scale.
>
> The reviewer points out the scale limitations of our experiments.  We have added a full 'Limitations' section discussing this point along with others.  While we generally expect the same effect to persist with scale, we cannot guarantee that the mitigations we proposed will continue to suffice at larger scale.  In particular, a systematic study of how block-scaling algorithm choice interacts with model size, and whether keeping a small set of sensitive computations in high precision remains sufficient to stabilize training when the bulk of forward and backward GEMMs are in MX low precision at $\ge$1B parameters, is an interesting direction for future work.
>
> 1. > Appendix D is titled "Differences Between our Proxy Model and LLM" but only covers attention and loss function ablations. Either broaden the discussion to cover the more fundamental structural gaps (outliers, embedding layers, vocabulary, real data distributions) or adjust the title.
>
> This is a valid concern, and we have substantially expanded Appendix D to discuss clear differences between our proxy model and LLM settings.  We have added a new subsection D.1 ("Where the Proxy Model and Language Model Training Runs Differ"), in which we tabulate the structural gaps between the two settings.  Table 3, in particular, lists missing ingredients such as causally masked attention, multiple norm types, etc.   We hope that these changes will let readers draw their own conclusions about the reliability of the proxy model, and potentially directions for more faithful representativeness.
>
> 4. > Appendix B. The noise model assumes linearized GD while experiments use Adam. The ||ζ_t||_op ≈ 2 divergence threshold is only established on the proxy.
>
> This is indeed correct, and we have updated the footnote in Appendix B to emphasize this difference.
>
> We thank the reviewer again for their comments to improve our manuscript and hope that the changes have addressed their remaining concerns.

---

### Review · Reviewer_yDuq · 2026-05-01

**Summary Of Contributions:**

The paper investigates training instabilities that arise when pretraining language models in Microscaling (MX) block-scaled precision formats. The authors make three main contributions:

1. Across nearly a thousand OLMo-style training runs spanning several compute budgets and a broad sweep over MX weight/activation precision combinations they document sharp, unrecoverable loss instabilities that become more pronounced at larger compute scales.
2. They introduce a residual MLP student–teacher proxy model that reproduces the failure mode. They identify the primary culprit as multiplicative gradient bias introduced when block-scale division clamps tightly clustered values to the largest representable number. The dominant contributors are the layer-norm affine weights.
3. They show that selective higher-precision treatment of layer-norm weights or activations, or quantizing only the forward pass, recovers performance competitive with full-precision training, and they provide scaling-law fits for the stabilized configurations.

**Strengths.** The combination of large-scale empirical sweeps and a controlled proxy model is well-executed; the proxy serves the paper's mechanistic goals without overclaiming. The overflow analysis is intuitive and well-supported. The intervention experiments cleanly establish causality.

**Weaknesses.** (i) The mechanism is centered on LayerNorm affine weights, but most modern LLMs (Llama, Qwen, OLMo-2, DeepSeek, Mistral) use RMSNorm; whether the clustering-and-overflow story carries over to the RMSNorm gain vector is not tested. (ii) The LLM sweep uses GeLU, but the authors' own synthetic results  flag SwiGLU as substantially more unstable — yet SwiGLU is the dominant activation in current open LLMs and is not part of the LLM sweep. (iii) (minor) Authors say "verify that this mechanism is not limited to synthetic settings but also emerges in the LM setting by evaluating mitigation strategies." Mitigation experiments provide *consistent* evidence but do not, on their own, verify a mechanism , the writing needs to be improved in that sentence.

**Audience:**

Yes

**Audience Explanation:**

MX formats are the precision target for the current generation of NVIDIA accelerators, and its important faster training of large models.

**Broader Impact Concerns:**

No broader impact concerns.

**Claims And Evidence:**

Yes

**Claims Explanation:**

The central empirical claims are well-supported and writing is clear in the most parts.

**Requested Changes:**

**1.  Address generalization to RMSNorm.** The paper's central mechanism centers on shared-scale overflow of LayerNorm affine weights. However, the dominant normalization in modern LLMs (Llama 3/4, Qwen, OLMo-2, DeepSeek, Mistral, etc.) is RMSNorm, which retains a learnable gain vector γ but drops the centering and bias. Because γ is still a learned per-channel scale, the same clustering-and-band-clamping mechanism could in principle apply — or it might not, if γ values are distributed differently. The paper currently does not test or discuss this. Please either (a) run a small RMSNorm variant of the LM and/or proxy experiments and report the γ overflow fractions analogous to Figure 3, or (b) at minimum add a discussion section addressing whether the proposed mechanism is expected to transfer to RMSNorm and why. This is important because, as written, a reader could reasonably question how broadly the prescription applies to current frontier models.

**2.Justify or extend the activation function choice in the LLM sweep.** The LLM experiments use GeLU, but the synthetic results (Figure 2a) and the cited Fishman et al. (2024) both indicate that SwiGLU is substantially more failure-prone in low precision. SwiGLU is also the dominant activation in modern open LLMs. Please either (a) repeat the key MXFP8 E4M3-E4M3 and E5M2-E5M2 LM sweeps with SwiGLU at a comparable compute budget, or (b) explicitly acknowledge this as a limitation in the main text and discuss the expected direction of the effect.

**3. (minor) Tighten the verification claim in the introduction.** The sentence "We verify that this mechanism is not limited to synthetic settings but also emerges in the LM setting by evaluating mitigation strategies" overstates the inference. Mitigation success is consistent with the proposed mechanism but does not, by itself, verify it. Please rephrase it.

---

> ### Author Response · Authors · 2026-05-25
> **Addressing Reviewer yDuq Feedback**
>
> We thank the reviewer for their valuable feedback.  We are glad that they found our results potentially useful to the broader community and for faster LLM training of larger models.
>
> We aim to address their requested changes as follows:
>
> # Weaknesses / Requested Changes
>
> > 1. Address generalization to RMSNorm. The paper's central mechanism centers on shared-scale overflow of LayerNorm affine weights. [...] Please either (a) run a small RMSNorm variant of the LM and/or proxy experiments and report the γ overflow fractions analogous to Figure 3, or (b) at minimum add a discussion section addressing whether the proposed mechanism is expected to transfer to RMSNorm and why.
>
> This is a valid point, as RMSNorm could in principle behave quite differently from LayerNorm.  While one may expect RMSNorm $\gamma$ to behave roughly the same as LayerNorm $\gamma$ during training, this is not immediately clear from the equations alone.
>
> To better understand this, we have updated Appendix E with a synthetic proxy experiment in which we replaced LayerNorm with RMSNorm, with all other configuration/hyperparameter settings held equal.  We find that RMSNorm $\gamma$ saturates just as fast as LayerNorm $\gamma$ (in fact, layer-0 RMSNorm $\gamma$ saturates faster and remains fully saturated for the rest of training), and that the activation overflow fractions are comparable between the two norms.  While this doesn't rule out the possibility that the LLM setting could behave differently, it is a suggestive prediction.
>
> > 2. Justify or extend the activation function choice in the LLM sweep. The LLM experiments use GeLU, but the synthetic results (Figure 2a) and the cited Fishman et al. (2024) both indicate that SwiGLU is substantially more failure-prone in low precision. SwiGLU is also the dominant activation in modern open LLMs. Please either (a) repeat the key MXFP8 E4M3-E4M3 and E5M2-E5M2 LM sweeps with SwiGLU at a comparable compute budget, or (b) explicitly acknowledge this as a limitation in the main text and discuss the expected direction of the effect.
>
> We have added a full Limitations section in the main body of the text discussing this point along with others.  In particular, we acknowledge that our LM sweep uses GeLU rather than the SwiGLU activation now common in LMs.  Our proxy results suggest SwiGLU is more failure-prone in low precision than GeLU or ReLU (removing LayerNorm stabilizes it only partially), so roughly speaking we would expect the LM to inherit the same effect. A full characterization of the interplay between activation functions and block-scaled precision is an interesting direction for future work.
>
> > 3. (minor) Tighten the verification claim in the introduction. The sentence "We verify that this mechanism is not limited to synthetic settings but also emerges in the LM setting by evaluating mitigation strategies" overstates the inference. Mitigation success is consistent with the proposed mechanism but does not, by itself, verify it. Please rephrase it.
>
> Our original sentence is indeed an overstatement of our claims.  We have edited this sentence to emphasize the consistency between the two settings, rather than as a causal verification of the mechanism:
>
> > This mechanism appears consistent with the failure modes observed in the LM setting, and we use this insight to design mitigations that stabilize LM training, including disabling layernorm quantization and using high precision in selective parts of the network computation.
>
> We thank the reviewer again for their comments to improve our manuscript.

---

### Review · Reviewer_z697 · 2026-05-15

**Summary Of Contributions:**

This paper investigates training instabilities when training LLMs using Microscaling (MX) low-precision formats (MXFP8, MXFP6) on NVIDIA Blackwell hardware. Across ~1000 language model runs, the authors document sharp, stochastic, and unrecoverable loss divergences. They develop a small residual MLP proxy model to isolate the failure mechanism, identifying the primary cause as overflow in layer-norm affine weight quantization, whose values cluster tightly during training, causing shared-scale clamping that biases gradients multiplicatively. They validate this with controlled intervention experiments and conclude with scaling law fits for hybrid precision configurations that recover near-bfloat16 performance.

Strengths:
- Large-scale sweep (~1000 models) gives strong empirical support.
- Root cause (layernorm affine overflow) is concrete and well-evidenced.
- Intervention experiments are rigorous: same seed/state, only precision changes.
- Practical hybrid configs are identified and validated with scaling laws.

Weakness:
- Figure 1 shows all ~70 runs overlaid for a single hyperparameter configuration (Table 3), but it is not discussed how this configuration was chosen or whether instability rates are consistent across other settings. Given that hyperparameter sensitivity is carefully characterized for the proxy model, a brief similar discussion for the LLM setting would be helpful.
- text overlaps page numbering on page 9.
- Proxy model validity is not quantified. It will be nice if authors can report false positive/negative rates (e.g. "when proxy was stable, LLM was stable X% of the time").
- Figure labeling can be improved. The paper alternates between proxy and LLM results without clear tags at certain places

**Audience:**

Yes

**Audience Explanation:**

Yes. Low-precision training is increasingly relevant as next-generation hardware (e.g. NVIDIA Blackwell) natively supports MX formats. Researchers and practitioners working on efficient LLM training will find the instability analysis and the proposed mitigations directly useful. The paper also contributes a reusable proxy model methodology and scaling law fits under various precision configurations, which are of broader interest to the ML efficiency and training dynamics communities.

**Broader Impact Concerns:**

No significant ethical concerns. The paper is a technical study of numerical precision in LLM training and does not involve human subjects, sensitive data, or dual-use risks.

**Claims And Evidence:**

Yes

**Claims Explanation:**

The claims are largely well supported. The instability phenomenon is observed across nearly one thousand models spanning a range of compute budgets and precision formats. The proposed mechanism (overflow of tightly clustered layernorm affine weights under shared-scale quantization) is well motivated and supported by the overflow fraction measurements in Figure 3. The intervention experiments in Section 5.2 strengthen the causal argument by isolating precision as the variable of interest. Scaling law fits provide supporting evidence for the proposed mitigations.

**Requested Changes:**

These will strengthen the work:

1. Proxy model validity should be better justified. The paper does not report how well the proxy predicts LLM stability. At minimum, a discussion of the cases where the proxy agrees or disagrees with the LLM setting would help the reader assess how much to trust proxy-based conclusions.

2. Figure 1 shows LLM results for a single hyperparameter configuration but does not explain how it was chosen or whether instability rates are consistent across other settings. Please provide a brief justification or additional hyperparameter sweep in the LLM setting.

3. It is also unclear how crashed runs (noted in footnote 1) were handled in the scaling law fits. This should be stated explicitly, as crashed runs may disproportionately represent unstable configurations and bias the fitted parameters.

4. Figures alternate between proxy and LLM results without consistent labeling. Adding explicit "Proxy" or "LLM" tag would improve readability.

5. Minor formatting issue: text overlaps with page numbering on page 9.

---

> ### Author Response · Authors · 2026-05-25
> **Addressing Reviewer z697 Feedback**
>
> We thank the reviewer for their insightful comments.  We are glad that they found the claims well supported, the small-scale proxy a potentially reusable methodology, and our work increasingly relevant for next-generation hardware and efficient LLM training.
>
> We have attempted to incorporate their requested changes as follows:
>
> # Weaknesses / Requested Changes
>
> 1. > Proxy model validity should be better justified. The paper does not report how well the proxy predicts LLM stability. At minimum, a discussion of the cases where the proxy agrees or disagrees with the LLM setting would help the reader assess how much to trust proxy-based conclusions.
>
> The reviewer notes that the paper is currently lacking information about the cases in which the proxy agrees or disagrees with the LLM setting.  To address this, we have significantly expanded Appendix D.  In particular, we have added a new subsection D.1 ("Where the Proxy Model and Language Model Training Runs Differ"), in which we emphasize that the proxy model is merely a probe and not a faithful model of an LLM.  We also note that we treat agreement between the proxy and the LLM as supporting evidence for a shared mechanism rather than as a causally predictive model. In this section, we explicate what the proxy model does and does not share with the LM setting, which interventions qualitatively agree between the two, and which ones were not tested.  We hope that these changes will let readers draw their own conclusions about the reliability of the proxy model, and potentially directions for more faithful representativeness.
>
> 2. > Figure 1 shows LLM results for a single hyperparameter configuration but does not explain how it was chosen or whether instability rates are consistent across other settings. Please provide a brief justification or additional hyperparameter sweep in the LLM setting.
>
> Thank you for raising this. We agree the original draft did not make our LR choice transparent. We have updated Appendix H to include our rationale for the learning rate choice.  In particular, we chose $\eta = 2 \times 10^{-4}$ as the largest peak LR at which every bf16 LM run we tested is stable. The choice is based on a coarse sweep over $\{1\mathrm{e}{-5},\,2\mathrm{e}{-4},\,1\mathrm{e}{-3}\}$ on the bf16 baseline.  We observed that 1e-3 sometimes causes instability in bf16 itself, 1e-5 is trivially stable due to a tiny parameter update but too small to be practical, and 2e-4 is the intermediate value that is reliably bf16-stable.
>
> 3. > It is also unclear how crashed runs (noted in footnote 1) were handled in the scaling law fits. This should be stated explicitly, as crashed runs may disproportionately represent unstable configurations and bias the fitted parameters.
>
> We have added a 'Limitations' section on page 12 to discuss this point, among others.  The crashed runs failed due to transient issues on our academic cluster, unrelated to the precision format itself, and were excluded from the scaling-law fits.  However, the reviewer's point is well taken in that this exclusion could in principle bias the fitted exponents, though we did not observe crashes correlated with model scale or precision format. We emphasize that the scaling-law fits reported here are intended as a sanity check that our hybrid configurations recover bfloat16-level scaling, rather than as precise estimates of the exponents.  A more careful analysis of block-scaled precision scaling laws, with confidence intervals on the fitted parameters, would indeed be an important direction for future work.
>
> 4. > Figures alternate between proxy and LLM results without consistent labeling. Adding explicit "Proxy" or "LLM" tag would improve readability.
>
> We have updated figures in the main body to clearly denote in the caption whether they are LLM or proxy model results, whenever there is potential ambiguity.
>
> 5. > Minor formatting issue: text overlaps with page numbering on page 9.
>
> We have fixed the page 9 formatting issue.
>
> We hope that with these changes we have addressed the reviewer's remaining concerns, and we thank them again for their feedback.

---

### Decision · Action_Editor_XHzK · 2026-07-04

**Recommendation:** Accept with minor revision

**Audience:**

Yes

**Audience Explanation:**

Yes. Low-precision training is increasingly relevant as next-generation hardware (e.g. NVIDIA Blackwell) natively supports MX formats. Researchers and practitioners working on efficient LLM training will find the instability analysis and the proposed mitigations directly useful. The paper also contributes a reusable proxy model methodology and scaling law fits under various precision configurations, which are of broader interest to the ML efficiency and training dynamics communities.

**Claims And Evidence:**

Yes

**Claims Explanation:**

The claims are generally well supported, especially after the revision given the constructive comments raised by the reviewers.

---

> ### Author Response · Authors · 2026-07-13
> **Camera ready version uploaded**
>
> Dear Action Editor,
>
> We thank you and the reviewers for the thoughtful assessment and constructive feedback on our paper. We have incorporated all of the reviewers' suggestions into the camera-ready draft, as well as author names, which we have now uploaded.
>
> We are grateful for the recognition of our work and for the feedback that has helped make it stronger.
>
> Sincerely,
> The Authors